# Entry, replication and innate immunity evasion of BANAL-236, a SARS-CoV-2-related bat virus, in *Rhinolophus* and human cells

Ségolène Gracias[1], Elodie Le Seac'h[1], Samuel Donaire-Carpio[1], Françoise Vuillier[2], Léa Vendramini[1,3], Adam Moundib[1], Sarah Temmam[3], Magdalena Rutkowska[4,5], Flora Donati[6], Anastasija Cupic[4,5], Javier Juste[7], Carles Martinez-Romero[4], Nathalie Morel[8], Olivier Schwartz[9], Nevan J. Krogan[10,11,12], Lisa Miorin[4,13], Marcel A. Müller[14,15], Caroline Demeret[2], Sandie Munier[16], Philippe Roingeard[17], Jyoti Batra[10,11,12], Adolfo Garcia-Sastre[4,13,18,19], Vincent Caval[1]*, Nolwenn Jouvenet[1]*

1 Institut Pasteur, Université Paris Cité, CNRS UMR 3569, Virus sensing and signaling Unit, Paris, France, 2 Institut Pasteur, Université Paris Cité, Interactomics, RNA and Immunity Unit, Paris, France, 3 Institut Pasteur, Université Paris Cité, Pathogen Discovery Laboratory, Paris, France, 4 Department of Microbiology, Icahn School of Medicine at Mount Sinai, New York, New York, United States of America, 5 Graduate School of Biomedical Sciences, Icahn School of Medicine at Mount Sinai, New York, New York, United States of America, 6 Institut Pasteur, National Reference Center for Respiratory Viruses, Paris, France, 7 Estación biológica de doñana, Avda, Seville, and CIBER Epidemiology and Public Health, CIBERESP, Madrid, Spain, 8 Département Médicaments et Technologies pour la Santé (DMTS), Service de Pharmacologie et Immunoanalyse, Université Paris Saclay, CEA, INRA, Gif-sur Yvette, France, 9 Institut Pasteur, Université Paris Cité, Virus and Immunity Unit, Paris, France, 10 Quantitative Biosciences Institute (QBI), University of California San Francisco, San Francisco, California United States of America, 11 Gladstone Institute of Data Science and Biotechnology, J David Gladstone Institutes, San Francisco, California, United States of America, 12 Department of Bioengineering and Therapeutic Sciences, University of California, San Francisco, California, United States of America, 13 Global Health Emerging Pathogens Institute, Icahn School of Medicine at Mount Sinai, New York, New York, United States of America, 14 Institute of Virology, Charité – Universitätsmedizin Berlin, Corporate Member of Freie Universität Berlin, Humboldt-Universität zu Berlin, and Berlin Institute of Health, Berlin, Germany, 15 German Center for Infection Research (DZIF), Partner Site Charité, Berlin, Germany, 16 Institut Pasteur, Université Paris Cité, Lyssavirus Epidemiology and Neuropathology Unit, Paris, France, 17 INSERM U1259 MAVIVH and INSERM US61, Université de Tours and CHU de Tours, Tours, France, 18 The Tisch Cancer Institute, Icahn School of Medicine at Mount Sinai, New York, New York, United States of America, 19 The Icahn Genomics Institute, Icahn School of Medicine at Mount Sinai, New York, New York, United States of America

* vincent.caval@pasteur.fr (VC); nolwenn.jouvenet@pasteur.fr (NJ)

## Abstract

Asian *Rhinolophus* bats are considered the natural reservoirs of an ancestral SARS-CoV-2. However, the biology of SARS-CoV-2-related viruses in bat cells is not well understood. Here, we investigated the replication of an isolate of BANAL-236, the only bat-derived SARS-CoV-2 relative isolated to date, in *Rhinolophus ferrumequinum* lungs (Rfe) cells. BANAL-236 did not replicate in wild-type *Rhinolophus* cell lines. Entry assays using pseudoviruses expressing the spike proteins (S) of SARS-CoV-2, BANAL-236, and BANAL-52 revealed that efficient S-mediated entry depends on the expression of human ACE2 (hACE2) and human TMPRSS2 (hTMPRSS2) in human and *Rhinolophus* cells. Through biochemical, virological, and electron microscopy analyses, we showed that BANAL-236 and SARS-CoV-2 completed their

**Data availability statement:** All relevant data are within the manuscript and its Supporting Information files. The sequencing data generated during the study have been deposited in the ArrayExpress database, under accession number E-MTAB-16878.

**Funding:** Studies performed in N.J's lab were supported by the Institut Pasteur, Centre National de la Recherche Scientifique (CNRS) and Agence Nationale de la recherche (ANR; grant number ANR-24-CE35-4716 EmerCoV). The O.S. lab is funded by Institut Pasteur, the Vaccine Research Institute (ANR-10-LABX-77), Labex IBEID (ANR-10-LABX-62-IBEID) and the European Union 4Health project DURABLE (grant 101102733). The Interactomics, RNA and Immunity Unit (C.D) received funding from the Task Force Covid19 of the Institut Pasteur to support this work. This research was also funded by grants from the National Institutes of Health to A.G-S (U19AI135972) and to N.J.K. and J.B (U19AI135990 and U19AI135972). The funders had no role in the study design, data collection and analysis, decision to publish, or preparation of the manuscript.

**Competing interests:** I have read the journal's policy and the authors of this manuscript have the following competing interests. The Krogan Laboratory has received research support from Vir Biotechnology, F. Hoffmann-La Roche, and Rezo Therapeutics. NJK has a financially compensated consulting agreement with Maze Therapeutics. NJK is the President and is on the Board of Directors of Rezo Therapeutics, and he is a shareholder in Tenaya Therapeutics, Maze Therapeutics, Rezo Therapeutics, and GEn1E Lifesciences.

replication cycles in RFe cells engineered to express high levels of hACE2 and hTM-PRSS2. Despite efficient viral replication in modified *Rhinolophus* and human cells, no induction of interferon (IFN)-stimulated genes was detected. Using a screening approach, we identified several BANAL-236 proteins that antagonize IFN production and signalling in human cells. Our findings thus show that BANAL-236 possesses critical features that enabled zoonotic spillover: hACE2 usage and potent evasion of human IFN responses. The *Rhinolophus* cellular model we established offers a platform for further investigating the interactions between bat sarbecoviruses and their reservoir hosts.

## Author summary

Bats are known reservoirs for viruses that cause severe diseases in humans, such as coronaviruses and filoviruses. Bat species naturally or experimentally infected with these viruses rarely exhibit clinical symptoms, suggesting an evolved tolerance to viral infections. To elucidate the mechanisms underlying viral tolerance and to identify factors that could facilitate zoonotic spillover, it is essential to study the replication of bat-borne viruses in relevant bat and human cellular models. Here, we investigated the replication of BANAL-236, a SARS-CoV-2 related virus isolated from fecal samples of *Rhinolophus* bats in Northen Laos, in a novel cell line derived from *Rhinolophus ferrumequinum* lung fibroblasts, as well as in human cells. Our findings reveal that BANAL-236 can efficiently use human entry factors and potently evade the human innate immune response, two traits that may have contributed to its zoonotic transmission. Furthermore, the *R. ferrumequinum* cell lines we developed is a valuable model for investigating the molecular interactions between coronaviruses and their natural hosts.

## Introduction

Severe acute respiratory syndrome coronavirus (SARS-CoV) and SARS-CoV-2 both belong to the subgenus *Sarbecoviruses* within the *Betacoronavirus* genus. They have the potential to cross species barriers, as demonstrated by their emergence in the human population in 2003 and 2019, respectively. In 2020, seven sarbecoviruses, collectively referred to as BANAL-CoVs, were recovered from fecal samples of *Rhinolophus* bats in Northern Laos [1]. Three of these viruses are among the closest known relatives of the Wuhan strain of SARS-CoV-2: *R. malayanus* BANAL-52, *R. pusillus* BANAL-103 and *R. marshalli* BANAL-236 [1]. An ancestor of SARS-CoV-2 may have originated from recombination and evolution of close relatives of these viruses and other *Rhinolophus* sarbecoviruses [1].

The spike (S) glycoprotein of sarbecoviruses mediates the attachment to and fusion with the host cell membrane. During its biosynthesis in the Golgi of infected cells, the S protein of SARS-CoV-2 is cleaved by proprotein convertases, such as

furin, into S1 and S2 subunits [2], which remain non-covalently associated on the surface of mature virions. This cleavage site is an exception within the Sarbecovirus subgenus, as the S protein of all other known members, including BANAL-CoVs, lack the furin-cleavage site (FCS) between the S1 and S2 subunits [1]. The receptor-binding domain (RBD) within the S1 subunit determines the binding affinity to the surface receptor angiotensin-converting enzyme 2 (ACE2) and is therefore a key determinant for host range and pathogenesis. Seventeen residues, so called 'contact residues' are essential for the RBD-ACE2 interaction [2]. The RBD of BANAL-236 differs from that of SARS-CoV-2 by only a single contact residue, yet it binds strongly to human ACE2 (hACE2) and facilitates hACE2-dependent entry and replication into human cells [1]. Binding of the S protein to ACE2 induces conformational changes in both subunits, exposing a S2 cleavage site called S2'. This cleavage is mediated by the transmembrane protease serine 2 (TMPRSS2) at the cell surface, triggering the formation of a fusion pore that releases into the cytosol the positive sense, single-stranded viral genome complexed with nucleocapsid (N) proteins [2,3]. In the absence of cell surface TMPRSS2, the virus enters cells via endocytosis and S2' is cleaved by endosomal proteases, such as cathepsin L [2,3]. BANAL-236 and BANAL-52 exhibit a greater dependence on TMPRSS2 than SARS-CoV-2 in Vero cells expressing both ACE2 and TMPRSS2 [4].

BANAL-236 was successfully isolated by inoculating rectal swabs of *R. marshalli* on trypsin-treated Vero-E6 cells [1], which are African green monkey kidney cells widely used for viral propagation. Replication in these cells was dependent on ACE2 [1]. BANAL-236 also replicated in human intestinal Caco-2 and Calu-3 cells, both of which express endogenous ACE2 [1]. Studies using infectious molecular clones of BANAL-236 and BANAL-52 showed that both viruses replicated less efficiently than SARS-CoV-2 in primary human nasal epithelial cells and in the upper airway of transgenic mice expressing hACE2, as well as in both the upper and lower airways of Syrian golden hamsters [4]. Furthermore, BANAL-236 and BANAL-52 exhibited reduced pathogenicity and pneumotropism compared to SARS-CoV-2 in hACE2-expressing mice and Syrian golden hamsters [4–6] and showed limited transmission among Syrian golden hamsters [4]. The attenuated replication and transmission of BANAL-CoVs cannot be attributed to poor affinity for host receptors, as BANAL-236 S protein binds both hamster and human ACE2 with higher affinity than the S protein of early SARS-CoV-2 isolates [1,5].

Mammalian cells have evolved robust antiviral defense mechanisms, including the interferon (IFN) response, to detect and control viral infections. The recognition of viral nucleic acids by immune sensors initiates a signaling cascade that culminates in the production and secretion of IFNs. These secreted IFNs bind to receptors on the surface of both infected and neighboring cells, activating the JAK/STAT pathway. This, in turn, induces the expression of approximately 2000 IFN-stimulated genes (ISGs) [7]. Many ISGs act to disrupt the viral life cycle by targeting specific stages of replication, such as viral entry, protein translation, genome replication or assembly of new virions. High-throughput loss- and gain-of-function screens in human cells have identified several human ISGs with potent antiviral activities against SARS-CoV-2 [8]. For instance, LY6E inhibits S-mediated fusion of SARS-CoV-2 and other coronaviruses in both human cells and mice [9], while the prenylated form of OAS1 activates the endoribonuclease RNase-L to degrade SARS-CoV-2 RNA [10]. The coordinated action of ISGs efficiently controls SARS-CoV-2 replication. Accordingly, impaired IFN responses in COVID-19 patients are associated with an increased risk of severe disease [11].

Like all viruses, sarbecoviruses have evolved strategies to evade the IFN response in human cells [12–15]. A substantial proportion of SARS-CoV and SARS-CoV-2 proteins are capable of inhibiting key components of IFN pathways [12,16–20]. For example, the viral proteases NSP3 and NSP5 cleave IRF3, thereby suppressing IFN production [21–23]. As a result, SARS-CoV-2 is a weak inducer of the IFN response in some human cell models, such as hACE2-expressing A549 lung cells and human lung tissues, compared to other respiratory RNA viruses [24–26]. In contrast, SARS-CoV-2 replication in bat cells derived from *Eptesicus serotinus, Eptesicus fuscus* and *Myotis myotis* expressing hACE2 triggers ISG expression [25,27], indicating that viral IFN counteraction mechanisms may be species-specific. However, the ability of BANAL-CoVs to counteract the IFN response in *Rhinolophus* cells remains unexplored.

Here, we investigated the replication of BANAL-236 in cells isolated from *Rhinolophus* bats, with a focus on viral entry mechanisms and the evasion of innate immunity.

## Results

### BANAL-236 and SARS-CoV-2 failed to replicate in RFe cells

To better understand the biology of BANAL-236 in *Rhinolophus*-derived cells, we first established a primary lung fibroblast cultures from the lung tissues of *a R. ferrumequinum* (RFe) bat collected in Spain [28]. The cells were then immortalized by expressing SV40-LT antigen [28]. To assess the expression of key viral entry factors, we performed RT-qPCR analysis to quantify endogenous ACE2 and TMPRSS2 mRNA levels in RFe cells, as well as in 293T and Caco-2 cells as comparisons. Consistent with previous reports [29], 293T and Caco-2 cells expressed moderate levels of ACE2, approximately $10^2$ copies per µg of total RNA (Fig 1A), whereas ACE2 mRNA was undetectable in RFe cells. Both 293T and RFe cells expressed less than 10 copies of TMPRSS2 per µg of total RNA. By contrast, Caco-2 cells expressed high levels of the protease, over $10^3$ copies per µg of total RNA (Fig 1A).

Next, we examined the replication kinetics of BANAL-236 and SARS-CoV-2 (Wuhan strain) in RFe cells. Caco-2 cells, which are susceptible to both viruses [1,5], served as positive controls. Based on pilot experiments, Caco-2 cells were infected with 100 more infectious particles of BANAL-236 than SARS-CoV-2 to achieve comparable viral RNA yields. RT-qPCR analysis revealed that SARS-CoV-2 RNA levels in Caco-2 cells increased until 48 hpi, reaching about $10^7$ genome copies per µg of total RNA and remained stable through 72 hpi (Fig 1B). The growth kinetics of BANAL-236 were slower than those of SARS-CoV-2 in Caco-2 cells, showing a less rapid increase in RNA yield from the first to the second time point (Fig 1C). Flow cytometry analyses were used to further evaluate viral replication. Around 50% of Caco-2 cells were positive for the SARS-CoV-2 N protein at 48 hpi (Fig 1D), while around 30% of Caco-2 cells were expressing the N protein of BANAL-236 at 72 hpi (Fig 1E). Thus, BANAL-236 replicated less efficiently than SARS-CoV-2 in Caco-2 cells, despite the higher inoculum, a finding that contrasts with a previous study reporting comparable replication of BANAL-236 and SARS-CoV-2 (WK-521 strain) in these cells [5].

In RFe cells, infection with SARS-CoV-2 or BANAL-236 at a MOI of 0.2 and 0.5, respectively, did not result in increased viral RNA levels over time, suggesting an absence of replication (Fig 1B-1C). In agreement with these data, no N positive cells were detected by flow cytometric analysis (Fig 1D-1E). The lack of viral replication may be due to the absence of endogenous ACE2 and/or a low TMPRSS2 (Fig 1A). To activate the S protein and allow viral fusion independently of TMPRSS2 and other proteases, viral input was treated with the serine-protease trypsin [29]. Pre-activation of SARS-CoV-2 or BANAL-236 with trypsin did not enhance viral RNA yield or the percentage of RFe cells expressing the N protein (Fig 1B-1E), suggesting that S protein cleavage is not the limiting factor for infection in RFe cells. The absence of viral RNA and protein production in RFe cells may therefore result from a lack of pro-viral factors, such as ACE2 (Fig 1A) and/or the presence of potent antiviral mechanisms.

### Efficient entry of BANAL-236 and BANAL-52 in RFe cells requires both hACE2 and hTMPRSS2

To investigate viral entry mechanisms and ensure reliable detection of human and *Rhinolophus* ACE2 and TMPRSS2, plasmids expressing tagged versions of the four proteins (r/hACE2-V5 and myc-h/rTMPRSS2) were produced. Since BANAL-236 and BANAL-52 were isolated from *R. marshalli* and *R. malayanus*, respectively [1] and the *R. marshalli* genome is unavailable, rACE2 was synthetized from *R. malayanus* sequence. Given the endogenous TMPRSS2 expression in RFe cells (Fig 1A), rTMPRSS2 was cloned from *R. ferrumequinum* cDNA to avoid having cells expressing TMPRSS2 from 2 species. The expression levels of the tagged proteins were assessed by western blot analysis in transfected 293T cells using antibodies against hACE2, hTMPRSS2, V5 or myc. Antibodies against hACE2 recognized as efficiently rACE2 and hACE2 (S1A Fig), while antibodies against hTMPRSS2 [30] did not detect its *Rhinolophus* ortholog (S1B Fig). Immunofluorescence experiments performed in transfected 293T cells using antibodies against hACE2 and/or myc revealed that the entry factors were well expressed (S1C Fig), consistent with our western blot analysis (S1A-S1B Fig). The 4 proteins localised to the cytoplasm. An overlap between the signals of the 2 specifies-specific pairs of proteins was observed, suggesting a distribution consistent with a potential interaction.

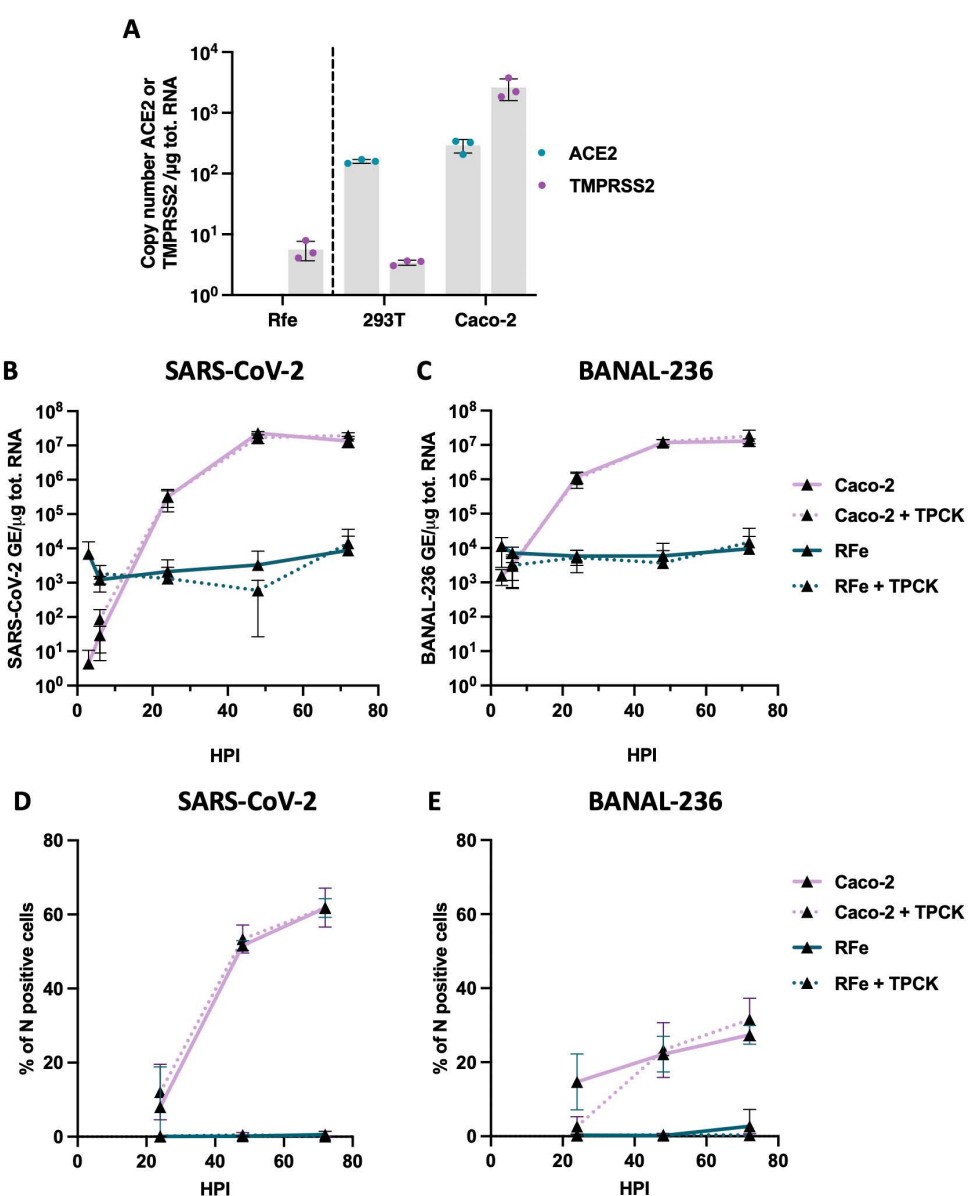

**Fig 1. SARS-CoV-2 and BANAL-236 replicate in Caco-2 cells, but not in RFe cells. (A)** Quantification of copy numbers per μg of total cellular RNA of endogenously expressed ACE2 and TMPRSS2 in indicated cell lines via RT-qPCR analysis. Data are means±SD of three independent experiments. Caco-2 cells (pink lines) were infected with SARS-CoV-2 **(B-D)** or BANAL-236 **(C-E)** at a multiplicity of infection (MOI) of 0.0002 and 0.02 respectively. Cells were treated or not with 1μg/ml of trypsin TPCK (dotted lines). RFe cells (dark green lines) were infected with SARS-CoV-2 **(B-D)** or BANAL-236 **(C-E)** at a MOI of 0.2 and 0.5 respectively and treated or not with 1μg/ml of trypsin TPCK (dotted lines). **(B-C)** The relative amounts of cell-associated viral RNA were measured by RT-qPCR analysis at different times post-infection and were expressed as genome equivalents (GE) per μg of total cellular RNAs. Data are the means±SD of three independent experiments. **(D-E)** The percentages of cells expressing the viral N protein were determined by flow cytometry analysis. Data are means±SD of three independent experiments.

To assess whether the V5 tag interferes with ACE2 receptor function, we performed SARS-CoV-2 infection experiments in human A549 lung cells. SARS-CoV-2 does not replicate in wild-type A549 cells unless they express hACE2 [25]. We established A549 cells stably expressing hACE2-V5 (S1D Fig). Flow cytometry analysis using an anti-S antibody at 24 hpi

(MOI 1) revealed that approximately 12% of hACE2-V5-expressing cells were positive for the S protein (S1E Fig), indicating that viral S protein binds hACE2-V5. Our previous experiments showed that around 3% of A549 expressing untagged hACE2 were expressing the SARS-CoV-2 S protein after 24 hours of infection with the same MOI [25]. Thus, the V5 tag does not interfere with hACE2 receptor function.

Since hACE2 is a known substrate of hTMPRSS2 [31,32], we performed Western blot analysis to determine whether myc-tagged hTMPRSS2 and rTMPRSS2 retain their ability to cleave hACE2. 293T cells were transfected with hACE2 and increasing amount of myc-TMPRSS2. In the absence of the protease, hACE2-V5 appeared as 2 bands of about ~125-kDa (S1F Fig), likely representing differentially glycosylated forms. In the presence of myc-hTMPRSS2, the heavier form of hACE2-V5 was not detectable anymore, while the ~120-kDa form remained visible. Additionally, a ~20-kDa ACE2 fragment appeared (S1F Fig), consistent with cleavage by myc-hTMPRSS2. These results align with previous studies performed in 293T cells expressing untagged hACE2 and hTMPRSS2 [31,32]. Identical hACE2 patterns were observed with myc-rTMPRSS2 (S1F Fig). Together, these results indicate that myc-tagged human and *Rhinolophus* TMPRSS2 retain protease activity and efficiently process hACE2.

To perform viral entry assays, we generated 293T cells stably expressing hACE2-V5 (hA), myc-hTMPRSS2 (hT), or both factors (hAT). Similarly, we generated 293T cells expressing rACE2-V5 (rA), myc-rTMPRSS2 (rT), or both factors (rAT). Flow cytometry analysis revealed that 20–40% of 293T cells were expressing hACE2-V5 (hA) at their surface (Fig 2A) while 40–80% were expressing myc-hTMPRSS2 (hT) at their surface (Fig 2B). The surface expression level of both factors was below the detection limit in wild-type cells (Fig 2A-2B). RT-qPCR analysis of the levels of exogenous hACE2 and hTMPRSS2 mRNAs in 293T cells were consistent with the flow cytometric analysis (S2A-S2B Fig). Flow cytometry showed that ~70% of 293T-rA cells expressed rACE2-V5 and ~40% of 293T-rAT expressed rACE2-V5 (Fig 2C). Around 40% of 293T-rT and 293T-rAT expressed myc-rTMPRSS2 (Fig 2D). RT-qPCR analysis of the abundance of rACE2 and rTMPRSS2 mRNAs in 293T cells were consistent with the flow cytometric analysis (S2D-E Fig). ACE2 expression in 293T cells was reduced in the presence of TMPRSS2, as compared to when it was expressed on its own (Figs 2A, 2C, S2A and S2D), likely due to the ability of the protease to cleave ACE2 (S1F Fig) [31,32].

Lentiviruses expressing the S proteins of SARS-CoV-2 (Wuhan strain), BANAL-236 or BANAL-52 were generated, as well as lentiviruses with no S, which served as negative controls. All three S proteins facilitated entry of pseudovirions into 293T-hACE2 cells (Fig 2E), but not into wild-type 293T. Expression of hTMPRSS2 alone was insufficient for efficient viral entry, although co-expression with hACE2 slightly enhanced S-mediated entry (Fig 2E). Expression of *Rhinolophus* entry factors, alone or in combination, did not support SARS-CoV-2 or BANAL-236 S-mediated entry in 293T cells (Fig 2E). Expression of rACE2 alone allowed entry of lentiviruses expressing the S proteins of BANAL-52 in 293T (Fig 2E), although not as efficiently as hACE2. This result was unexpected given that BANAL-52 was originally detected in *R. malayanus*. When myc-rTMPRSS2 was expressed together with rACE2-V5 in 293T cells, S-mediated entry of BANAL-52 was less efficient than in 293T cells expressing rACE2 alone (Fig 2E). This is likely due to the lower expression of rACE2 in 293T-rAT cells compared to 293T-rA cells (Figs 2C and S2D).

To investigate viral entry mechanisms in another cellular models, experiments were conducted in hamster BHK-21 kidney cells, which are commonly used for these assays [33]. Cells were transfected with the tagged versions of the human and *Rhinolophus* entry factors and then transduced with S-bearing lentiviruses (S2E Fig). Pilot experiments conducted with plasmids expressing GFP revealed that only a small percentage (around 5%) of cells were GFP positive. This discrepancy may explain why luciferase activity was approximately 2 logs lower in BHK-21 cells compared to 293T cells (S2E Fig). Despite a low transfection efficiency, the results showed that expression of hACE2 in BHK-21 cells promoted efficient entry of lentiviruses bearing the S proteins of all 3 viruses (S2E Fig), consistent with the data obtained in 293T cells (Fig 2E). However, unlike in 293T cells (Fig 2E), co-expression of hTMPRSS2 with hACE2 in BHK-21 cells did not enhance the entry of the lentiviruses, as compared to cells expressing hACE2 alone (S2E Fig). Additionally, expression of rACE2 alone did not facilitate entry of lentiviruses expressing the S protein of BANAL-52 in BHK-21 cells. These results

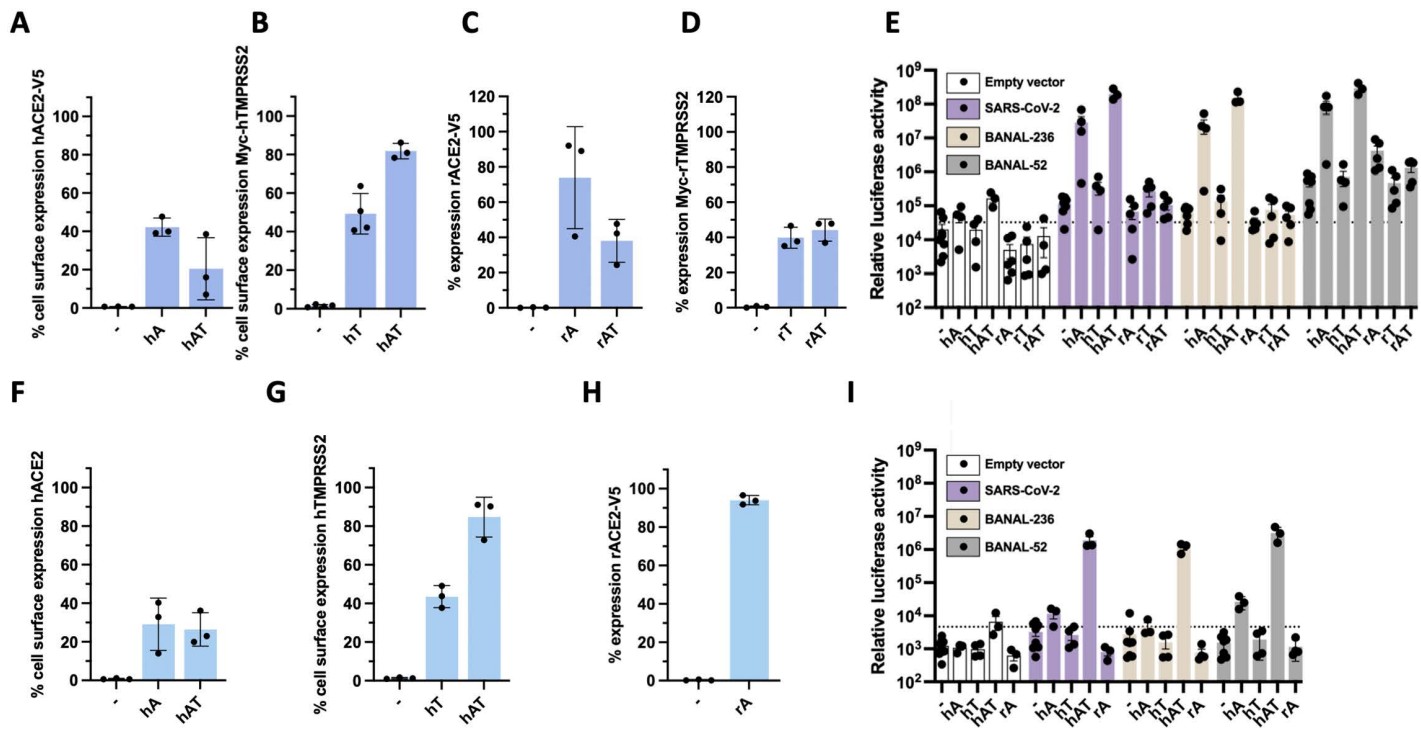

**Fig 2. Pseudovirus assays revealed that efficient entry of BANAL-236 and BANAL-52 in RFe cells requires both hACE2 and hTMPRSS2. (A-B)** Cell surface expression of hACE2-V5 **(A)** and myc-hTMPRSS2 **(B)** were determined by flow cytometry analysis prior to the entry assays shown in **(E)** in wt 293T cells (-) or expressing hACE2-V5 **(hA)**, Myc-hTMPRSS2 (hT) or both entry factors (hAT). Data are the means±SD of three independent experiments. **(C-D)**. Cell surface expression of rACE2-V5 **(C)** or Myc-rTMPRSS2 **(D)** were determined by flow cytometry analysis prior to the entry assays shown in **(E)** in wt 293T cells (-) or expressing rACE2-V5 **(rA)**, myc-rTMPRSS2 (rT) or both entry factors (hAT). Data are the means±SD of three independent experiments. **(E)** Entry assays were performed in wt 293T cells (-) or stably expressing hACE2-V5 **(hA)**, myc-hTMPRSS2 **(hT)**, rACE2-V5 **(rA)**, myc-rTMPRSS2 **(rT)**, or both entry factors (hAT or rAT). Cells were transduced with the same HIV-1 p24 quantity of pseudo-lentiviruses bearing the S proteins of SARS-CoV-2 (purple), BANAL-236 (beige) or BANAL-52 (grey). Results are expressed in relative luminescence units (RLU). The dashed line indicates the average RLU obtained with cells transduced with empty vectors (white). Data are means±SEM of three independent experiments. **(F-H)** Cell surface expression of hACE2 **(F)**, hTMPRSS2 **(G)** or rACE2-V5 **(H)** were determined by flow cytometry analysis prior to the entry assays shown in (I) in wt RFe cells (-) or expressing hACE2 **(hA)**, hTMPRSS2 **(hT)**, both entry factors (hAT) or rACE2-V5. Data are the means±SD. **(I)** Entry assays were performed in wt RFe cells (-) or stably expressing hACE2 **(hA)**, hTMPRSS2 **(hT)**, both human entry factors (hAT), or rACE2-V5 **(rA)**. Cells were transduced with the same HIV-1 p24 quantity of pseudo-lentiviruses bearing the S proteins of SARS-CoV-2 (purple), BANAL-236 (beige) or BANAL-52 (grey). Results are expressed in relative luminescence units (RLU). The dashed line indicates the average RLU obtained with cells transduced with empty vectors (white). Data are the means±SEM of three independent experiments.

are likely due to a low transfection (and possibly transduction) efficiencies in BHK-21 cells. Nevertheless, despite the limitations of this cellular model, its use revealed that expression of hACE2 is necessary and sufficient to promote the entry of lentiviruses expressing the S proteins of the 3 sarbecoviruses in 293T and BHK-21 cells. Of note, the inability of rACE2-V5 to mediate the entry of lentiviruses bearing the SARS-CoV-2 S protein in 293T (Fig 2E) and BHK-21 cells (S2E Fig) explained why expression of rACE2-V5 did not support SARS-CoV-2 replication in A549 cells (S1D-S1E Fig).

Finally, to perform entry assays in *Rhinolophus* cells, we generated RFe cells stably expressing hACE2, rACE2, hTMPRSS2 or both human factors. Unfortunately, repeated attempts to produce RFe cells stably expressing rTMPRSS2, alone or in combination with rACE2, were unsuccessful. Untagged versions of the human entry factors were used in these experiments. Flow cytometric analysis revealed that approximately 30% of RFe-hA and RFe-hAT cells expressed hACE2 at their surface (Fig 2F), despite RFe-hAT cells expressing slightly higher levels of hACE2 mRNA than RFe-hA cells (S2F Fig). Around 40% of RFe-hT and 90% of RFe-hAT expressed hTMPRSS2 at their surface (Fig 2G), despite exhibiting

similar abundance of hTMPRSS2 mRNA (S2G Fig), suggesting a more efficient plasma membrane addressing of hTM-PRSS2 in RFe-hAT than in RFe-hT. Finaly, about 90% of RFe-rACE2 cells expressed rACE2-V5 (Figs 2H and S2H). Expression of hACE2 or hTMPRSS2 alone in RFe cells did not support efficient viral entry (Fig 2I). Similarly, no viral entry was observed in RFe-rA cells (Fig 2I), suggesting that expression of rACE2 alone is insufficient to mediate S-dependent entry in RFe cells, as observed in 293T cells (Fig 2E). Only co-expression of both human entry factors enabled efficient viral entry in RFe cells (Fig 2I). These results suggest that the undetectable levels of ACE2 and low TMPRSS2 levels in RFe cells (Fig 1A) contribute to the absence of viral replication.

Given that RFe-hAT cells supported efficient entry of lentiviruses pseudotyped with the BANAL-236 S protein (Fig 2I), we selected them for further investigation.

## BANAL-236 and SARS-CoV-2 replicate in RFe-ATC cells

We assessed the ability of BANAL-236, and, for comparison, SARS-CoV-2, to replicate in RFe-AT cells. Entry assays showed that these cells supported entry of lentiviruses bearing the S proteins of both viruses (Fig 2I). Despite maintaining RFe cells under antibiotic selection, the expression of hACE2 and hTMPRSS2 was unstable. Therefore, the cell surface expression of hACE2 and hTMPRSS2 was evaluated during each experiment by flow cytometry (Fig 3A). Around 20% of RFe-AT cells were positive for hACE2 and 50% for hTMPRSS2 (Fig 3A). RFe-AT cells were infected with SARS-CoV-2 or BANAL-236 at a MOI of 0.2. RT-qPCR analysis revealed that viral RNA levels did not increase over time (Fig 3B-3C), suggesting an absence of viral replication. Although not significant, a modest increase of BANAL-236 and SARS-CoV-2 RNA was observed at late time post-infection in RFe-AT cells treated with TPCK compared to untreated cells (Fig 3B-3C), suggesting that trypsin activation of the S protein may enable a small fraction of viruses to enter cells. Consistent with the RT-qPCR results (Fig 3B-3C), flow cytometry detected no N protein expression in RFe-AT cells, even in the presence of trypsin (Fig 3D-3E). This suggests that any viruses that entered the cells were blocked at an early stage of the replication cycle. Collectively, these data demonstrate that, although hACE2 and hTMPRSS2 expression permits S-mediated entry in RFe-AT cells (Fig 2I), it is insufficient to support viral replication. Of note, SARS-CoV-2 rapidly adapted to replicate in RFe-AT cells by acquiring a single point mutation in N [28].

Due to the unstable expression of hACE2 and hTMPRSS2 in RFe-AT cells, we produced over 30 clonal cell lines. Among these, only one clone (RFe-ATC) exhibited a cytopathic effect following SARS-CoV-2 infection. Flow cytometry analysis revealed that approximately 25% of RFe-ATC cells expressed hACE2 at their surface, a proportion comparable to that observed in RFe-AT cells (Fig 3A). In contrast, a higher proportion of RFe-ATC cells (around 90%) expressed hTMPRSS2 at their surface (Fig 3A), despite similar overall expression levels (Fig 3F). RFe-ATC cells were infected with SARS-CoV-2 or BANAL-236 at a MOI of 0.2 and 0.5, respectively. RT-qPCR analyzes showed that SARS-CoV-2 RNA levels increased by approximately 1-log during the first 20 hours of infection, followed by a plateau (Fig 3B). In contrast, BANAL-236 RNA levels rose steadily between 20- and 72-hours post-infection in RFe-ATC cells (Fig 3C). Flow cytometry analysis showed that a small percentage of RFe-ATC cells were positive for SARS-CoV-2 N protein at 24 hpi, increasing to 60% at 72 hpi (Fig 3D). Consistently with the viral RNA yield data (Fig 3C), BANAL-236 replication in RFe-ATC cells was delayed compared to SARS-CoV-2 (Fig 3E). Taken together, these results demonstrate that both SARS-CoV-2 and BANAL-236 are capable of replicating in RFe-ATC cells.

To determine whether RFe parental and RFe-AT were generally resistant to viral infection, we infected them with Vesicular Stomatitis virus (VSV), a negative-strand RNA virus from the *Rhabdoviridae* family that is known for its broad cell tropism [34]. Flow cytometry analysis using an antibody against the VSV G protein revealed that VSV replicated similarly in RFe, RFe-AT and RFe-ATC cells (Fig 3G), suggesting that the inability of SARS-CoV-2 and BANAL-236 to replicate in RFe and RFe-AT cells is not due to a pre-existing antiviral state.

The high surface expression of hTMPRSS2 in RFe-ATC cells (Fig 3A) may underlie their susceptibility to SARS-CoV-2 and BANAL-236. To test this hypothesis, we used siRNAs to knock-down hTMPRSS2 expression, selecting a

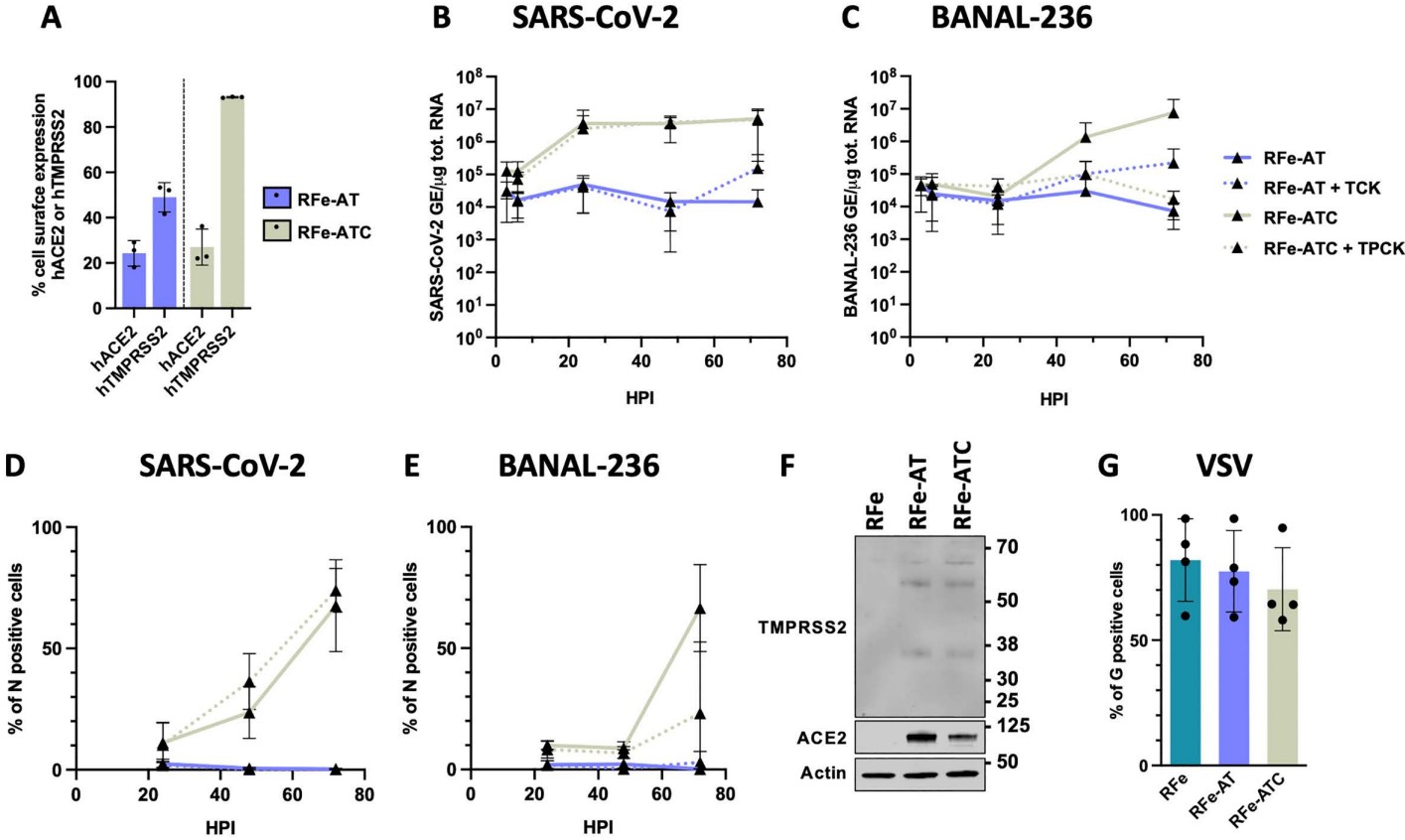

**Fig 3. RFe-ATC cells, but not RFe cells, are permissive to SARS-CoV-2 and BANAL-236. (A)** Expression of hACE2 at the cell surface of RFe-hACE2-hTMPRSS2 cells (RFe-AT, in blue) or clonal cells (RFe-ATC, in light green) were determined by flow cytometry analysis. Data are means±SD of three independent experiments. RFe-AT (blue lines) and RFe-ATC (green lines) cells were infected with SARS-CoV-2 **(B-D)** or BANAL-236 **(C-E)** at a multiplicity of infection (MOI) of 0.2 and 0.5, treated or not with 1μg/ml of trypsin TPCK (dotted lines). Cells were collected at the indicated hours post-infection (HPI). **(B-C)** The relative amounts of cell-associated viral RNA were measured by RT-qPCR analysis and were expressed as genome equivalents (GE) per μg of total cellular RNAs. Data are the means±SD of three independent experiments. **(D-E)** The percentages of cells expressing the viral N protein were determined by flow cytometry analysis. Data are means±SD of three independent experiments. **(F)** hACE2 and hTMPRSS2 expression in RFe, RFe-AT, and RFe-ATC cells. Whole-cell lysates were analysed by western blotting with antibodies against the indicated proteins. Data are representative of two independent experiments. **(G)** RFe, RFe-AT and RFe-ATC cells were infected with VSV at a MOI of 0.05 for 16 hours. The percentages of cells expressing the viral G protein were determined by flow cytometry analysis. Data are means±SD of three independent experiments.

concentration that reduced the proportion of surfaced hTMPRSS2-positive RFe-ATC cells (Fig 4A), to levels comparable to those in RFe-AT cells (Fig 3A). Under these conditions, BANAL-236 and BANAL-52 S-mediated entry was modestly, but reproducibly, less efficient in RFe-ATC cells expressing reduced levels of surface hTMPRSS2, as compared to cells transfected with control siRNAs (Fig 4B). Viral replication in the presence of reduced levels of hTMPRSS2 was then assessed by flow cytometry by measuring the number of N protein-positive cells at 72 hpi. Viral replication was completely abolished in these cells, as compared to cells treated with control siRNAs (Fig 4C), suggesting a strong dependence on hTMPRSS2. This pronounced effect likely reflects the cumulative impact of reduced hTMPRSS2 expression over multiple rounds of replication, in contrast to the modest effect observed during a single-entry event. Thus, the susceptibility of the RFe-ATC cells to viral infection may be attributed to their high level of surface hTMPRSS2 expression, as compared to RFe-AT cells.

PLOS Pathogens

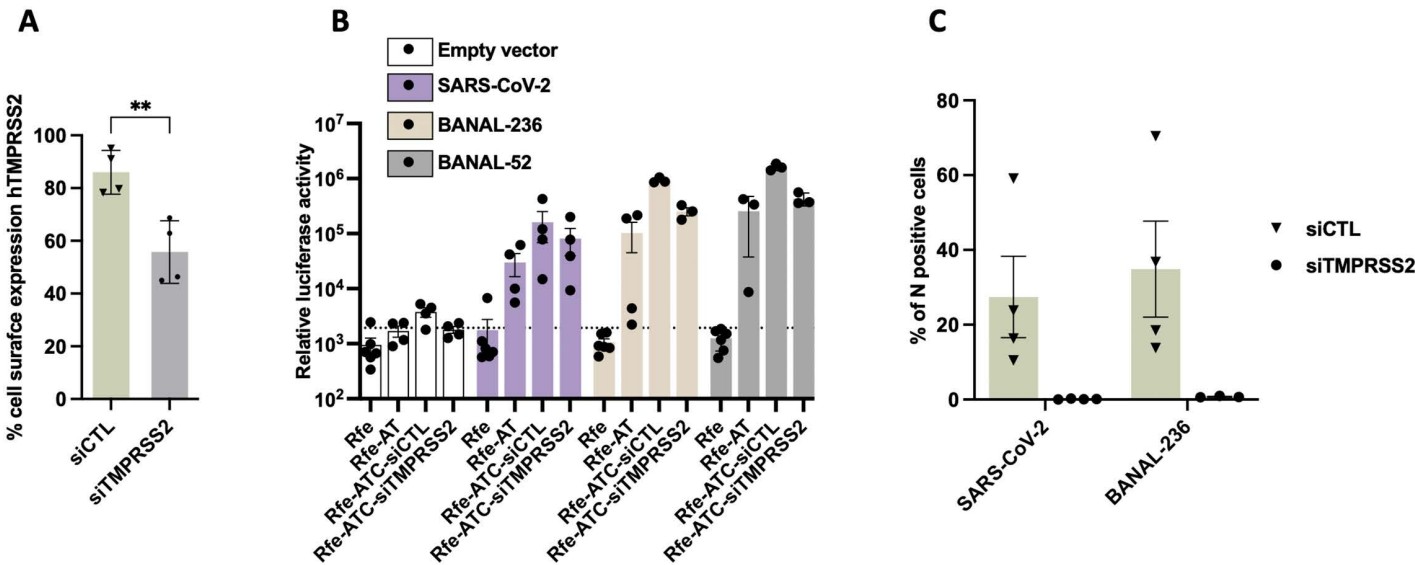

**Fig 4. BANAL-236 replication in RFe-ATC cells is dependent on high cell surface TMPRSS2 expression. (A)** RFe-ATC cells were transfected with siRNAs targeting TMPRSS2 or control siRNAs (siCTL) for 3 days. The percentages of cells expressing hTMPRSS2 at their surface were determined by flow cytometry analysis. Data are the means±SD of four independent experiments. T-tests two-tailed were performed **: p<0.01. **(B)** Entry assays were performed with lentiviruses pseudotyped with the S proteins of SARS-CoV-2 (purple), BANAL-236 (beige) or BANAL-52 (grey). Cells were transduced with the same HIV-1 p24 quantity of pseudo-lentiviruses. Results are expressed in relative luminescence units (RLU). The dashed line indicates the average RLU obtained with cells transduced with empty vectors. **(C)** Cells were infected with SARS-CoV-2 and BANAL-236 at a multiplicity of infection (MOI) of 0.02 and 0.2, respectively. Seventy-two hours later, cells were collected and the percentages of cells expressing the viral N protein were determined by flow cytometry analysis. Data are means±SD of at least three independent experiments.

## SARS-CoV-2 and BANAL-236 complete their replication cycles in RFe-ATC cells

To further characterize SARS-CoV-2 and BANAL-236 replication in RFe-ATC cells, we performed transmission electron microscopy on cells infected for 72 h, with Caco-2 cells included for comparison and non-infected cells as negative controls (S3 Fig). In infected Caco-2 cells, we observed numerous viral replication factories composed of double-membrane vesicles (DMVs) (Fig 5A-5B), as previously described in SARS-CoV-2 infected Vero cells [35], although the DMVs in these cells displayed slight structural differences consistent with variations reported across different cell lines [36]. These DMVs, which likely derive from ER membranes and autophagic processes, serve as sites for viral RNA synthesis [36,37]. Fully assembled virus particles were detected within large intracellular vesicles (Fig 5C), some of which fused with the plasma membrane (Fig 5D), suggesting virion release via exocytosis. Many virions remained attached to the surface of infected cells (Fig 5D), possibly due to high hACE2 expression at the plasma membrane. RFe-ATC cells infected with SARS-CoV-2 exhibited similar ultrastructural changes (Fig 5E-H), including numerous DMVs. Large vacuoles containing virions (Fig 5G) and surface-bound virions (Fig 5H) were also present in RFe-ATC cells. In contrast, Caco-2 and RFe-ATC cells infected with BANAL-236 contained very few DMVs (Fig 5I-5P), with only occasional DMV-like structures observed (Fig 5N, white arrows). In addition, large vacuoles filled with dozens of BANAL-236 virions were observed in both cell lines (Fig 5J and 5O), some of which were connected to the plasma membrane (Fig 5P). The presence of virions at the surface of RFe-ATC cells (Fig 5N and 5P) suggested that BANAL-236 completed its replication cycle in these bat cell line.

To assess the infectivity of virions observed at the cell surface, supernatants from infected RFe-ATC and Caco-2 cells were titrated on Vero-E6 cells (Fig 5Q-R). Supernatants from Caco-2 cells contained approximately $5.10^5$ TCID$_{50}$/mL of infectious BANAL-236 or SARS-CoV-2 particles (Fig 5Q). Similarly, about $10^5$ TCID$_{50}$/mL were retrieved from the

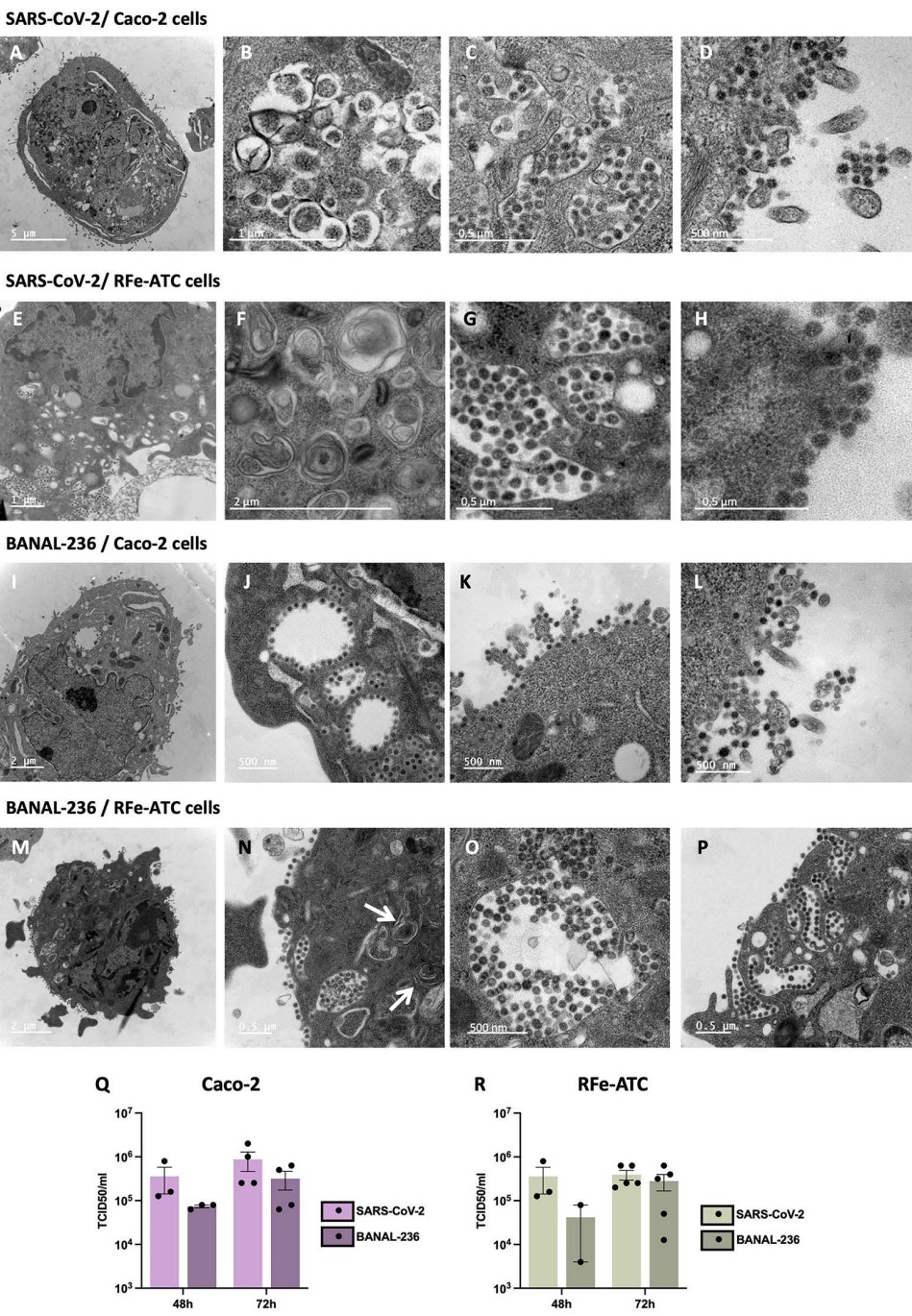

**Fig 5. SARS-CoV-2 and BANAL-236 complete their replication cycle in RFe-ATC cells. (A-P)** Caco-2 and RFe-ATC cells were infected for 72 **h.** Caco-2 cells were infected with SARS-CoV-2 **(A-D)** or BANAL-236 **(I-L)** at an MOI 0.001 and 0.01 respectively. RFe-ATC cells were infected with SARS-CoV-2 **(E-H)** and BANAL-236 **(M-P)** at MOI 0.02 and 0.2 respectively. Cells were then subjected to transmission electron microscopy analysis. DMV-like structures are pointed by white arrows. **(Q-R)** At 48 and 72 h post-infection supernatants from Caco-2 **(Q)** and RFe-ATC **(R)** cells were collected. Titration of clarified supernatants by $TCID_{50}$ assays was performed on Vero-E6 cells infected for 5 days. Data are the means±SEM of at least two independent experiments.

supernatant of infected RFe-ATC cells (Fig 5R). These results confirmed that both viruses complete their replication cycles in RFe-ATC cells.

## BANAL-236 antagonizes the IFN response in *Rhinolophus* and human cells

Given the critical role of viral IFN antagonism in overcoming species barrier, we evaluated ISG induction in RFe cells and their derivatives following infection with BANAL-236 or SARS-CoV-2. We compared mRNA abundance of *OAS1* and *ISG20*, two ISGs that are conserved across vertebrate species [38], in infected cells. No upregulation of *OAS1* or *ISG20* expression was observed in RFe nor RFe-AT cells at 72 hpi (Fig 6A-6B), which was consistent with the absence of viral replication in these cells (Figs 1 and 3). Despite robust replication of BANAL-236 and SARS-CoV-2 in Caco-2 and RFe-ATC cells (Figs 1 and 3), no induction of *OAS1* or *ISG20* expression was observed either at 72 hpi (Fig 6A-6B). Similar results were obtained when the mRNA abundance of these two ISGs were evaluated in cells infected for 4, 24, and 48 hours (S4 Fig). These results suggest that both viruses efficiently counteract the IFN response in these cell lines.

To ensure that RFe cells were immunocompetent, we first assessed ISG induction upon stimulation with polyI:C, a synthetic dsRNA analog. The mRNA levels of *OAS1* and *ISG20* increased upon stimulation in all four cell lines (Fig 6A-6B), suggesting intact IFN-induction and -signaling pathways. We then conducted a transcriptomic analysis of RFe-AT and RFe-ATC cells following stimulation with polyI:C. The analysis identified 181 differentially expressed genes (DEGs) (fold change ≥1.5 log2, p-value < 0.05) in stimulated RFe-AT cells compared to unstimulated control cells, and 166 DEGs in stimulated ATC cells (Figs 6C-6D and S5A, and S1 Table). A substantial proportion of the upregulated genes were shared between both cell lines, including ISGs that are conserved among mammalian species (such as *BST2*, *MX1*, and *IRF*s) (S5A Fig and S1 Table). Notably, *OAS1* and *ISG20* were among the most highly upregulated genes (Fig 6C-6D). To identify pathways affected by poly(I:C) treatment, we performed Gene Ontology term analysis on the upregulated genes. As expected, both cell lines exhibited significant enrichment in pathways related to viral infection and IFN responses (Fig 6E-6F). These findings indicate that both cell lines are similarly immunocompetent. Finally, we infected both RFe-ATC and RFe-AT cells with Japanese Encephalitis Virus (JEV), an orthoflavivirus known to circulate in multiple bat species, including *Rhinolophus* species [39]. Cells were infected with JEV at an MOI of 10, and flow cytometric analysis revealed that approximately 30% of cells were positive for the viral E protein at 24 hpi (S5B Fig). RT-qPCR analysis showed a modest, yet reproducible, increase in the mRNA levels of *OAS1* and *ISG20* following JEV infection in both cell lines (S5C-S5D Fig). This modest increase, compared to that observed in polyI:C-treated cells (Fig 6A-6B), is likely due to the ability of JEV to counteract the IFN response [40,41]. Together, these results provide evidence that both cell lines exhibited similar immunocompetence following a viral infection or polyI:C treatment. Thus, the absence of IFN induction during BANAL-236 replication is not due to a dysregulation of the IFN response in RFE-ATC cells, but rather reflects the virus's ability to actively counteract this response.

To investigate the ability of the 2 viruses to antagonize IFN signalling in individual cells, we assessed IFITM3 expression, another ISG that is conserved across vertebrate species [38], using fluorescent microscopic assays in Caco-2 cells (due to the lack of suitable antibodies for *Rhinolophus* ISGs). IFITM3 was detected in about 10% of unstimulated Caco-2 cells, but in approximately 45% of cells treated with IFNα2 (Fig 6G-6H). In cells infected for 72 hours, IFITM3 was detected in fewer than 5% of N-positive cells (Fig 6G-6H). When infected cells were treated with IFNα2 16 hours before fixation (Fig 6G-6H), fewer than 10% of cells were positive for both N and IFITM3, confirming that SARS-CoV-2 or BANAL-236 replication limits ISG expression, as observed in the RT-qPCR analysis (Fig 6A-6B).

To identify BANAL-236 proteins capable of antagonizing the IFN response in human cells, the open reading frames (ORFs) of all 30 viral proteins were cloned downstream of a N-terminal 3X-FLAG tag and expressed in 293T cells, which are easy-to-transfect cells. Western blot analyses confirmed the expression of 27 BANAL-236 proteins at their expected sizes (Fig 7A), including NSP3, the largest coronavirus protein (~220 kDa). A second ~80 kDa NSP3 band was detected, as previously reported for SARS-CoV-2 NSP3 [19]. NSP3 is composed of around 15 domains, including the papain-like

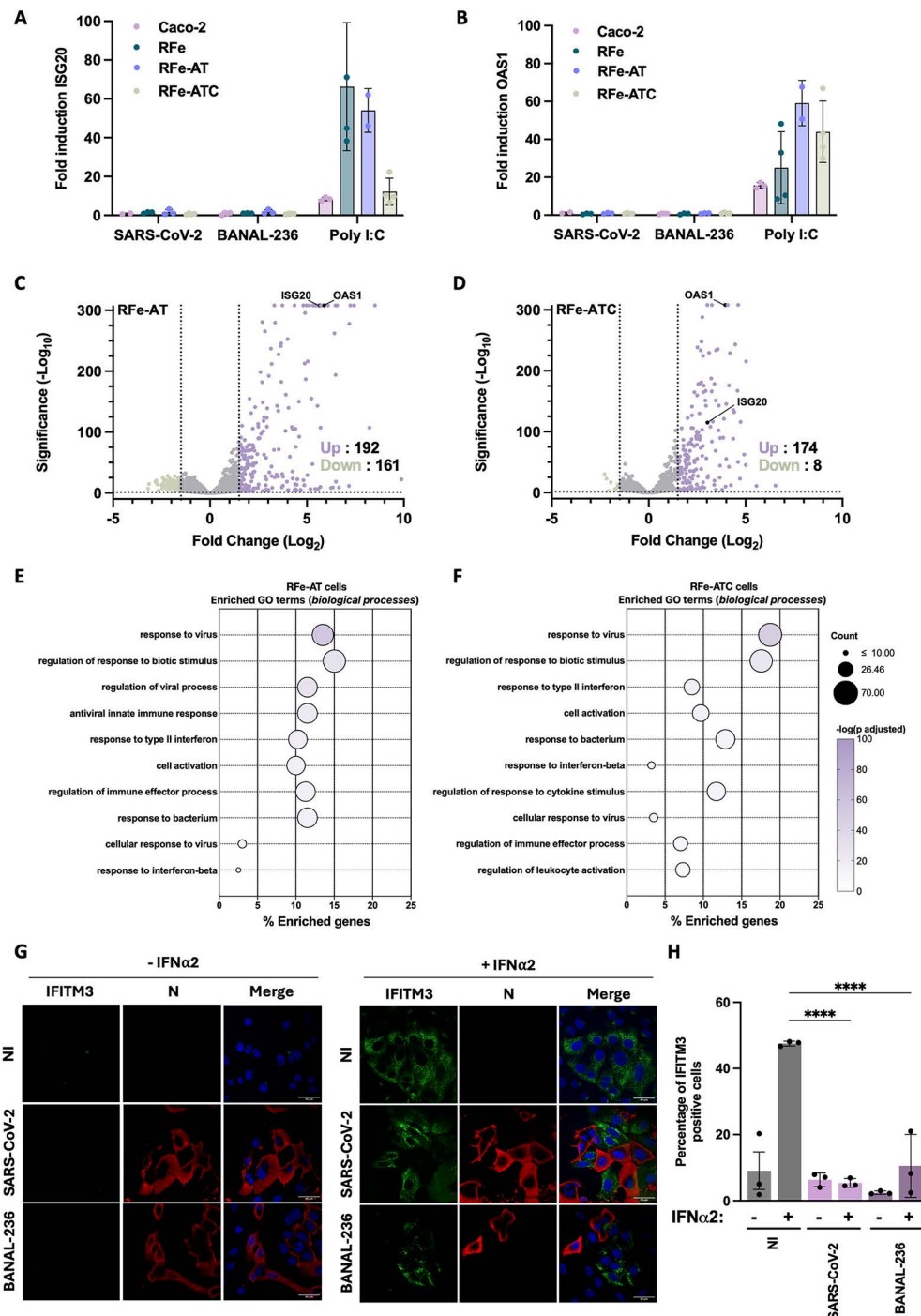

**Fig 6. SARS-CoV-2 and BANAL-236 replication failed to induce the expression of Interferon-stimulated genes in Caco-2 and RFe-ATC cells.**
**(A-B)** Caco-2 (pink), RFe (dark green), RFe-AT (blue) and RFe-ATC (grey) cells were infected with SARS-CoV-2 or BANAL-236 at a multiplicity of infection (MOI) of 0.2 and 0.5, respectively, for 72 hours. Alternatively, cells were treated with 100-500 ng of poly I:C for 16 hours. The relative amounts of *ISG20* **(A)** and *OAS1* **(B)** mRNAs were determined by RT-qPCR analysis. Results were first normalized to GAPDH mRNA and then to mRNA levels of control cells (mock-treated or non-treated cells), which were set at 1. Data are means ± SD of at least two independent experiments. **(C-D)** Volcano plots representing differentially expressed genes in RFe-AT (C) and RFe-ATC (D) cells upon polyI:C treatment (100 ng for 16 hours) compared to mock (PBS) transfected cells. Differentially expressed genes were selected using a log2 fold-change > 1.5 with and adjusted pvalue ≤ 0.05. **(E-F)** Top 10 enriched GO terms for Biological Process ranked by the adjusted p-value, for upregulated genes identified in RNA-Seq comparison comparing poly

I:C treated cells to mock (PBS) transfected condition for RFe-AT **(E)** and RFe-ATC **(F)**. Genes were defined as upregulated if the log2 fold change was equal or above 1.5 with a significance of adjusted pvalue below 0.05 compared to mock (PBS) transfected condition. **(G)** Caco-2 cells were infected for 72 hours with SARS-CoV-2 or BANAL-236 at a MOI of 0.0002 and 0.02, respectively, or left non-infected (NI). They were stimulated with IFNα2 at 1 000 IU/ml (lower panel) or mock-treated (upper panel) 16 hours before fixation. They were stained with antibodies recognizing IFITM3 (green) and the viral nucleocapsid N (red). Nucleus were counterstained using Hoechst dye (blue). Images are representative of three independent experiments. Scale bars, 40 μm. **(H)** Percentages of IFITM3-positive cells among cells expressing the viral N proteins were estimated by analysing at least 40 cells per condition. For uninfected (NI) cells, at least 60 cells were randomly picked. Data are mean ± SD. One-way ANOVA tests with Dunnett's correction were performed. ****$P < 0.0001$.

protease [42], and this extra band could represent an autoproteolytic product. FLAG-ORF8 was poorly expressed, and FLAG-S was unproperly processed. We thus generated ORF8 and S sequences with a C-terminal STREP-tag. ORF8-STREP was detected at the correct size (Fig 7A). Both the uncleaved precursor and the dissociated S2 fragment of S-STREP were detected, as expected [43]. Due to its small size (1.5 kDa), NSP11 was undetectable and was thus excluded from the analysis.

The effect of BANAL-236 proteins on the activation of the IFN-stimulated response elements (ISREs) was investigated in 293T cells overexpressing a constitutively active form of RIG-I (ΔRIG-I), a potent inducer of IFN production [44]. ISREs can be activated directly by transcription factors of the IFN induction pathway, like IRF3, in an IFN-independent manner [45]. Cells transfected with an empty plasmid (EV) served as references. Cells expressing a FLAG-tagged version of the NS1 of influenza A virus (IAV), a known IFN antagonist [46], served as positive controls. Its expression indeed reduced ISRE activation by over 100-fold (Fig 7B). TBEV NS5, which inhibits the JAK-STAT pathway [47], had no effect on ISRE activation (Fig 7B), confirming that ISRE was activated in an IFN-independent manner in our experimental settings. Four BANAL-236 proteins diminished the activity of the ISRE by at least 10-fold: NSP1, NSP6, NSP13, and ORF6 (Fig 7B), mirroring the antagonistic effects of their SARS-CoV-2 counterparts detected in similar experiments [16,17,20], despite sequence differences (S2 Table). To determine whether BANAL-236 proteins could inhibit IFN-I signalling, we repeated the screen in 293T cells stimulated with IFNα2 (Fig 7C). As expected [47], TBEV NS5 inhibited ISRE activation by around 10-fold (Fig 7C). NSP1, NSP3, NSP15 and S modestly suppressed the JAK-STAT pathway (Fig 7C). By contrast, SARS-CoV-2 expresses several proteins that potently block ISRE activation in IFN-I treated 293T cells, such as NSP1, NSP6, NSP13, NSP14, ORF6 and ORF7b [16,17,19,20]. Together, these findings showed that BANAL-236 is armed with mechanisms to evade the IFN response in human cells, albeit with a potential narrower repertoire of IFN-I signalling antagonists compared to SARS-CoV-2.

Since NSP1, NSP13 and ORF6 were the most efficient at inhibiting the IFN response in 293T cells (Fig 7B), we tested their impact on IFN response in RFe cells. Pilot experiments performed with plasmids encoding for GFP and several transfection reagents showed that the highest percentage of GFP-positive cells achieved was only 5%, which is insufficient to reliably assess the impact of viral proteins on IFN responses in these cells. We thus generated RFe cells stably expressing FLAG-NSP1, FLAG-NSP13, FLAG-ORF6, or GFP as a control. Western blots analysis of stable RFe cells revealed that NSP13 was expressed at a decent level, but ORF6 expression was low (Fig 7D). Most cells expressing NSP1 died, which is consistent with its known ability to block translation [12], and these cells were thus excluded from the analysis. RFe cells stably expressing FLAG-NSP13, FLAG-ORF6, or GFP were stimulated with polyI:C or left untreated for 16 hours. RT-qPCR analysis showed that the relative amounts of *ISG20* and *OAS1* mRNAs were significantly lower in RFe cells expressing FLAG-NSP13, compared to control cells transduced with empty lentiviruses (mock) or lentiviruses expressing GFP (Fig 7E-7F). These results, combined with the ones obtained with 293T cells expressing the BANAL-236 proteins (Fig 7B), suggest that NSP13 inhibits the IFN response in human and *Rhinolophus* cells. The inability of ORF6 to modulate ISG expression in RFe cells is likely due to its low level of expression.

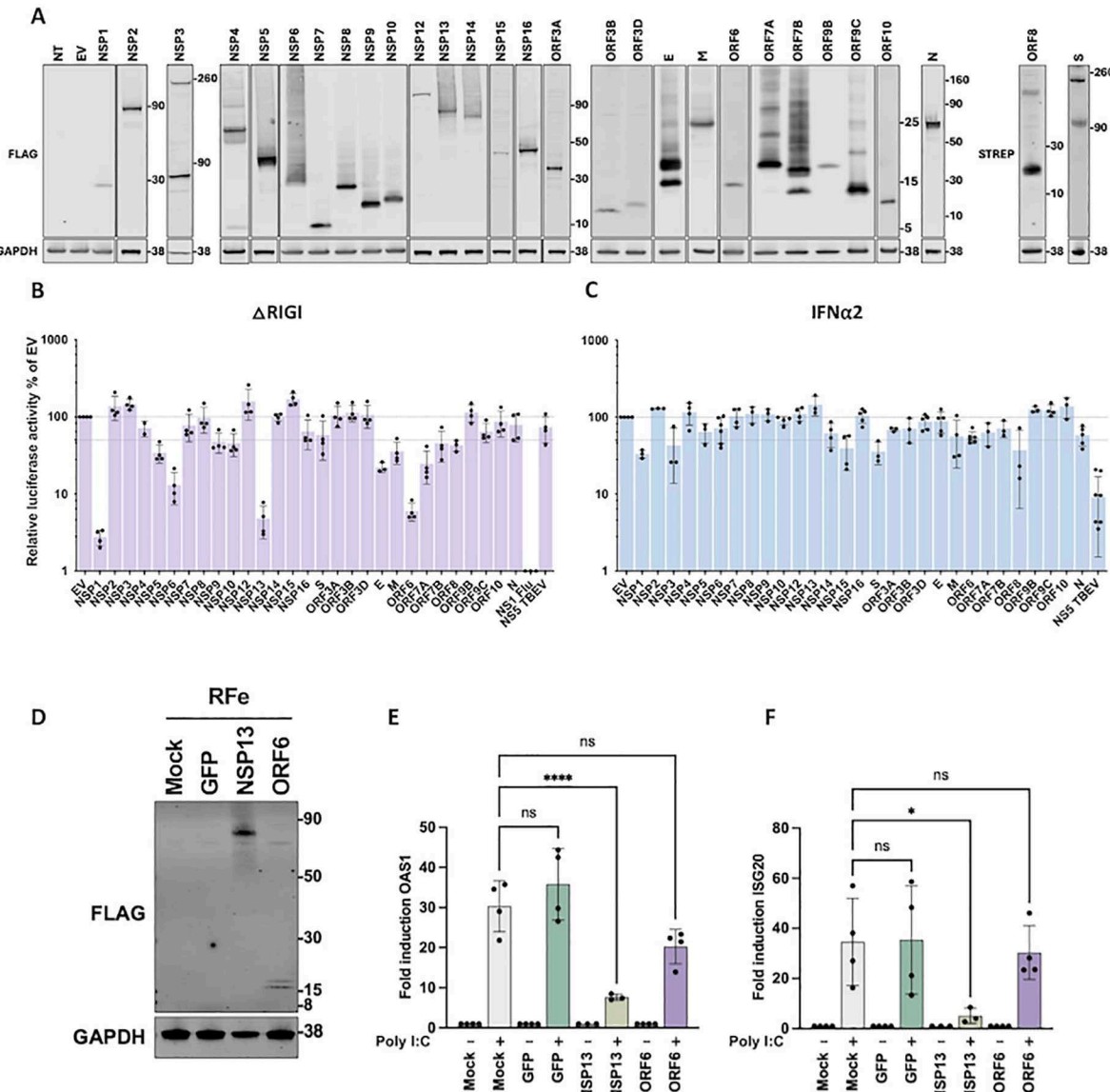

**Fig 7. BANAL-236 antagonizes the IFN response in human cells. (A)** Viral protein expression in 293T cells. 293T cells were mock-transfected (NT), transfected with empty plasmids (EV) or with plasmids encoding FLAG or STREP-tagged BANAL-236 viral proteins. Cells were harvested 24 h post-transfection and protein expression was assessed by Western blotting with anti-FLAG, anti-STREP and anti-GAPDH antibodies. Data are representative of three biological replicates. **(B)** 293T cells were co-transfected for 24h with the Firefly luciferase reporter plasmid p-ISRE-luc, TK Renilla luciferase control plasmid phRluc-TK, △RIGI plasmid and plasmids encoding individual viral proteins of BANAL-236. Empty vectors (EV) and plasmids encoding the NS1 of Flu protein were used as negative and positive controls, respectively. The data were analyzed by first normalizing the Firefly luciferase activity to the Renilla luciferase (Rluc) activity and then to EV samples, which were set at 100%. Data are means±SD of three biological replicates. One-way ANOVA tests with Dunnett's correction were performed compared to EV non-significant are not shown, *P<0.1, **P<0.01, ***P<0.001, ****P<0.0001 **(C)** 293T cells were co-transfected with the Firefly luciferase reporter plasmid p-ISRE-luc, TK Renilla luciferase control plasmid phRluc-TK, and plasmids encoding individual viral proteins of BANAL-236. Empty vectors (EV) and plasmids encoding the NS5 of TBEV protein were used as negative and positive controls, respectively. Cells were stimulated 7 h post-transfection with IFNα2 at 200 IU/ml and assayed for luciferase activity 16 h later. The data were analysed by first normalizing the Firefly luciferase activity to the Renilla luciferase (Rluc) activity and then to EV samples, which were set at 100%. Data are means±SD of three biological replicates. One-way ANOVA tests with Dunnett's correction were performed compared to EV non-significant are not shown, *P<0.1, **P<0.01, ****P<0.0001. **(D)** RFe cells stably expressing BANAL-236 NSP13 and ORF6 proteins upon lentiviral transduction. Cells transduced with empty lentiviruses (mock) or lentiviruses expressing GFP were used as negative controls. Whole-cell lysates were analyzed by western blotting with antibodies against the indicated proteins. Data are representative of two independent experiments. **(E-F)** The relative amounts of *ISG20* **(E)** and *OAS1* **(F)** mRNAs levels determined by RT-qPCR analysis after poly I:C treatment for 16 hours. Results were first normalized to GAPDH mRNA

and then to mRNA levels of mock (PBS) transfected control cells. Data are means±SD of at least three biological replicates. One-way ANOVA tests with Tukey's correction were performed. ns: non-significant, * $P < 0.1$, **** $P < 0.0001$.

## Discussion

We established a fibroblast cell line from the lung tissue of *R. ferrumequinum* (RFe) [28], a bat species widely distributed across Europe, Northern Africa, Central Asia and Eastern Asia. These cells were refractory to infection with BANAL-236 and SARS-CoV-2 replication. Similarly, primary fibroblast-like cells derived from the patagium of *R. ferrumequinum* were also non-permissive to SARS-CoV-2 replication [25], while lung and kidney cells from *R. sinicus* likewise failed to support SARS-CoV-2 replication [48,49], indicating that multiple *Rhinolophus*-derived cell lines are refractory to sarbecovirus replication.

RFe cells did not allow entry of lentiviruses pseudotyped with the S protein of either of these two sarbecoviruses, or of BANAL-52. This resistance aligns with the absence of ACE2 expression and low levels of TMPRSS2 in these cells. RFe-AT cells, which stably expressed hACE2 and hTMPRSS2, were also resistant to viral replication, despite supporting entry of S-bearing lentiviruses. Overcoming the entry blockade in RFe-AT cells was thus insufficient to permit viral replication, suggesting that post-entry mechanisms, such as the absence of essential pro-viral factors and/or the presence of potent antiviral factors, inhibit the replication of sarbecoviruses in these cells. Only one clonal cell line derived from RFe-AT cells (RFe-ATC) was permissive to BANAL-236 and SARS-CoV-2. A high proportion of these cells expressed hTMPRSS2 at their surface. Our results suggest that this elevated hTMPRSS2 surface expression may underlie their permissiveness, as even a modest reduction in hTMPRSS2 levels in RFe-ATC cells abolished viral replication over 3 replication cycles. This is consistent with studies in human cells, where TMPRSS2-mediated cell surface fusion enhances SARS-CoV-2 entry and downstream events, such as viral replication, transcription, and release [50]. Our data suggest a similar role for TMPRSS2 in facilitating BANAL-236 replication in RFe-ATC cells. Additional differences between RFe-AT and RFe-ATC cells, such as variations in basal expression of IFNs or antiviral ISGs, may further explain their distinct permissiveness to viral replication.

The receptor-binding domain (RBD) of the S protein of sarbecoviruses is a critical determinant of ACE2 binding and, consequently, viral host range. Among the 17 contact residues essential for ACE2 interaction [51–53], the RBDs of BANAL-52 and -236 differ from that of SARS-CoV-2 by only one (H498Q) or two (K493Q and H498Q) residues, respectively [1]. Given this high sequence similarity, we anticipated that these viruses would use ACE2 from their natural hosts and closely related species. However, S-mediated entry of BANAL-236 and BANAL-52 was more efficient in 293T cells expressing hACE2 than in 293T cells expressing comparable levels of *R. malayanus* ACE2. This trend extended to a broader panel of *Rhinolophus* ACE2 orthologs: entry of lentiviruses carrying the S proteins of BANAL-236 and BANAL-52 was more efficient in 293T expressing hACE2 than in 293T cells expressing ACE2 from 7 other *Rhinolophus* species (*R. affinis*, *R. cornutus*, *R. ferrumequinum*, *R. macrotis*, *R. pearsonii*, *R. pusillus*, *R. shameli*, and *R. sinicus*) [54]. Although ACE2 expression levels were not quantified in these experiments [54], these results, together with ours, suggest hACE2 supports entry of the S proteins of BANAL-236 and BANAL-52 more efficiently than ACE2 from at least 8 *Rhinolophus* species, including *R. malayanus*, the species from which the BANAL-52 sequence was recovered [1]. That said, the results of the entry assays must be interpreted cautiously. Our study used only one *R. malayanus* ACE2 sequence, yet *Rhinolophus* ACE2 alleles are known to exhibit substantial variabilities in their ability to support viral entry [55–57]. Furthermore, current sequencing efforts have not exhaustively sampled the full diversity of *Rhinolophus* ACE2 alleles. Finaly, BANAL-52 may circulate in multiple *Rhinolophus* species and its reservoir host could be a species other than *R. malayanus*. Supporting this possibility, lentiviruses bearing the BANAL-52 S protein entered 293T cells more efficient when expressing ACE2 from *R. macrotis*, *R. shameli* and *R. sinicus*, compared to cells expressing the other 4 tested *Rhinolophus* ACE2 [54]. One of these 3 species could thus be the natural reservoir for BANAL-52. To further clarify the host

range of BANAL-related coronaviruses, future studies should systematically evaluate the binding affinity of their RBDs to ACE2 from a comprehensive collection of alleles from different *Rhinolophus* species. Such approach would enable definitive conclusions about host range and ACE2 usage. Of note, lentivirus-based pseudotyping may not fully recapitulate the cell entry dynamics of bat sarbecovirus [58]. Therefore, using an alternative VSV-based system could provide additional insights into viral entry mechanisms.

Our findings reveal that BANAL-236 replicated less efficiently than SARS-CoV-2 in RFe-ATC cells. This attenuated replication aligns with previous reports in human primary lung cells and in hamster lungs [4]. The lower replication rate is not due to poor receptor affinity: BANAL-236's S protein binds hamster and human ACE2 with even higher affinity than the S protein of early SARS-CoV-2 isolates [1,5]. The attenuated growth and transmission of BANAL-CoVs, as compared to SARS-CoV-2, may partially result from the absence of an FCS. This hypothesis is supported by studies showing that SARS-CoV-2-ΔFCS exhibits reduced pathogenicity in hACE2-mice and hamsters when intranasally inoculated [5,59], and lower transmissibility in ferrets, compared to wild-type SARS-CoV-2 [60]. However, since BANAL-236 replicates even less efficiently than SARS-CoV-2-ΔFCS in human iPSC-derived airway epithelial cells and hamsters [5], the lack of an FCS is not the sole determinant of its attenuated growth in human respiratory cells. Additionally, the low percentage of N-positive RFe-ATC cells observed up to 48 hpi, suggests that BANAL-236 may have acquired adaptive mutations to enhance its replication after two rounds of infection.

BANAL-236 was isolated from bat feces, suggesting it may be an enterotropic virus in *Rhinolophus* bats [1]. Consistently, in macaques infected *via* the nasal and tracheal routes simultaneously, BANAL-236 behaved like an enteric virus [6]. Aditionally, BANAL-236 replicated more efficiently than SARS-CoV-2 in iPSC-derived human colon organoids, particularly in colonocytes [5]. Similar to SARS-CoV and SARS-CoV-2 in humans [61–63], BANAL-236 may infect both the respiratory and gastrointestinal tracts of *Rhinolophus* bats. Thus, further investigations are needed to study BANAL-CoVs replication in bat cells derived from multiple tissues, including gastrointestinal tissues, particularly from Southeast Asian *Rhinolophus* species, such as *R. marshalli* and *R.malayanus.*

Kinetic experiments performed in Caco-2 or RFe-ATC cells revealed that BANAL-236 replication failed to induce *ISG20* and *OAS1,* despite responding well to polyI:C stimulation. Consistently, BANAL-236 and BANAL-52 infection triggered only modest ISG induction in primary human bronchial and nasal epithelial cells [4]. These results suggest that both viruses suppress IFN responses across several cellular models. Additionally, BANAL-236 and BANAL-52 were resistant to IFN treatment in Vero-hACE2-hTMPRSS2 cells [4], indicating that they have evolved potent mechanisms to evade immune response in diverse mammalian cells. This is not surprising since SARS-CoV-2 efficiently counteracts IFN response through many of its proteins [16,20] and that BANAL-236 shares about 99.96% nucleotide sequence identity with SARS-CoV-2 (S2 Table). We identified four BANAL-236 proteins (NSP1, NSP6, NSP13, and ORF6) that potently antagonize the IFN induction pathway in human cells. Their SARS-CoV-2 counterparts are known to be IFN antagonists [16,17,19,20]. Assessing the IFN-evasion capacity of these proteins in RFe cells proved challenging due to their low transfection efficiency with multiple reagents (Lipofectamine, Lipojet, Jetprime and Genejammer). Nevertheless, we managed to generate RFe cells stably expressing decent level of NSP13 and showed that this viral protein effectively dampened IFN response in these cells. While BANAL-CoVs likely never encountered human cells, they have evolved to evade IFN responses in their natural bat hosts and the components that they target are probably highly conserved across mammalian species. For example, SARS-CoV-2 NSP1 and NSP13 prevent the phosphorylation of human IRF3 and TBK1 respectively [64,65]. BANAL-236 NSP1 and NSP3 likely employ similar mechanisms to target IRF3 in *Rhinolophus* cells and, consequently, in human cells. Our findings suggest that BANAL proteins more efficiently antagonize components of the IFN induction pathway than those of the JAK/STAT pathway, possibly because the JAK/STAT pathway is more divergent between humans and bats.

In summary, our findings show that BANAL-236, the likely ancestor of SARS-CoV-2, possesses key features that facilitated zoonotic spillover: its entry can be mediated by human entry factors and it efficiently evades human IFN responses.

Additionally, the RFe-ATC cells we developed provide a valuable new model for studying molecular interactions between sarbecoviruses and their natural hosts.

## Materials and methods

### Cell lines

Human colorectal adenocarcinoma Caco-2 cells (kind gift from Nathalie Sauvonnet, Institut Pasteur, Paris), African green monkey kidney epithelial Vero-E6 cells (ATCC CRL-1586), A549 lung cancer cells (ATCC CCL-185), and human embryonic kidney (HEK) 293T cells (ATCC CRL-3216), hereinafter referred to as 293T, were maintained in Dulbecco's Modified Eagle Medium (DMEM) (Gibco) containing GlutaMAX I, sodium pyruvate (Invitrogen) supplemented with 10% heat-inactivated fetal bovine serum (FBS) (Dutscher) and 1% penicillin and streptomycin (10 000 IU/ml; Thermo Fisher Scientific). BHK-21 golden hamster kidney fibroblasts were cultivated in Eagle's Minimum Essential Medium (EMEM) (Gibco) supplemented with 10% heat-inactivated FBS (Dutscher), 1% penicillin and streptomycin (10 000 IU/ml; Thermo Fisher Scientific) and 1x non-essential amino acids. The *Rhinolophus ferrumequinum* cells (RFe) were derived from lung tissue of a bat captured in Jerez, in Southern Spain. Primary lung fibroblasts were immortalized using lentiviruses expressing the Simian Vacuolating Virus 40 large T antigen (SV40T) [28]. RFe cells were cultured in medium containing 50% DMEM with GlutaMAX I, sodium pyruvate (Invitrogen) and 50% F-12 (Ham) medium (Gibco), supplemented with 10% heat-inactivated fetal bovine serum (FBS) (Dutscher) and 1% penicillin and streptomycin (10.000 IU/ml; Thermo Fisher Scientific). All cells were maintained at 37°C in a humidified atmosphere with 5% $CO_2$.

RFe and 293T cell lines stably expressing hACE2, hTMRSS2, rACE2-V5 or C-myc-rTMPRSS2 were generated by lentiviral transduction. $2.10^5$ cells were seeded in 6-well plates. Cells were transduced with 300 µL of lentivirus in 1 ml of medium containing 1x Polybrene the following day. After 3 days of incubation at 37°C, cells were either directly used for experiments or placed under antibiotic selection (6 µg/ml of puromycin and/or 6 µg/ml of blasticidin for RFe cells; 1 µg/ml of puromycin and 10 µg/ml of blasticidin for 293T cells). RFe cells stably expressing hACE2 and hTMRSS2 (RFe-AT) were selected with 5 µg/ml blasticidin and 250 µg/ml hygromycin (Invitrogen). RFe-AT cells were used to derivate clones (RFe-ATC) using limiting dilution.

### Viruses and infections

Experiments with BANAL-236 and SARS-CoV-2 viruses were conducted in a Biosafety Level 3 (BSL-3) laboratory, following safety protocols approved by the Institut Pasteur's risk prevention service. The SARS-CoV-2 strain BetaCoV/France/IDF0372/2020 and BANAL-236 were obtained from the National Reference Center for Respiratory Viruses hosted by Institut Pasteur. Viral stocks were produced by amplification on Vero-E6 cells, for 72 h in DMEM supplemented with 2% FBS and 1% P/S.Cells were exposed to the virus for 3 hours in a low volume of FBS-free medium. Cells were infected with the viruses at different MOIs as indicated in each figure For infection in the presence of TPCK-trypsin, it was added after 3 hours of infection, at a concentration of 1 µg/ml, in the medium described above or in DMEM containing 2% FBS.

Experiments with Japanese encephalitis virus (JEV) strain UVE/JEV/UNK/TW/RP9 (kind gift from Philippe Desprès, UMR PIMIT, Reunion Island) were conducted in a Biosafety Level 3 (BSL-3) laboratory, following safety protocols approved by the Institut Pasteur's risk prevention service. Cells were exposed to the virus for 2 hours in a low volume of 2% FBS containing medium. RFe-AT and RFe ATC cells were infected at a MOI of 10. The percentage of infection was assessed 24 hours post infection using the 4G2 antibody, an pan-flavivirus antibody raised against the viral envelope (E) protein by flow-cytometry.

Experiments with Vesicular stomatitis virus (VSV) and lentiviruses were performed in a BSL-2 + setting following biosafety regulations of the Institut Pasteur, Paris. The VSV Indiana strain was kindly provided by N. Escriou (Institut Pasteur).

## TCID$_{50}$ assays

Supernatants of infected cells were first cleared of cell debris by centrifugation at 500 rpm for 5 minutes at 4°C. They were 10-fold serially diluted in DMEM supplemented with 2% FBS and 1% P/S. Vero-E6 cells in suspension were infected with 10-fold serial dilutions in 96-well plates. Cells were seeded at a concentration of 80 000 cells/ml and incubated for 5 days in DMEM medium containing 2% FBS. At the end of the incubation period, cells were washed twice with PBS and then fixed with a crystal violet solution containing 3% formaldehyde for 30 minutes at room temperature. Cytopathic effects (CPE) were assessed by calculating the 50% tissue culture infective dose (TCID$_{50}$) using the Spearman-Karber method [66].

## Cloning of host proteins and production of pseudoviruses

The sequence of *Rhinolphus malayanus* ACE2 (rACE2) was synthesized with a C-terminal V5-6 His sequence tag and was amplified using primers designed to add a 5' BclI restriction site (rACE2 for: 5'-TCTAGTTGATCAGCCACCATGT CAGGCTCTAGCTGGC) and 3' XhoI restriction site (rACE2-V5 rev: 5'-GAGAGGCTCGAGTCTATCAATGGTGATGGTG). Human ACE (hACE2) coding sequence was amplified from pLenti6-hACE2 previously described [67] with the same primer design strategy as for rACE2 (hACE2 for: 5'-TCTAGTTGATCAGCCACCATGTCAAGCTCTTCCTGGCTCCTTC, hACE2 rev: 5'- GAGAGGCTCGAGTCGAGTTAAAAGGAGGTCTGAACATCATCA). PCR products were digested using BclI/XhoI restriction enzymes and cloned in the pFlap-Ubc-nLuc-IRES-Puro lentiviral vector (kind gift from Pierre Charneau, Institut Pasteur, Paris, France) digested with BamHI/XhoI.

Human TMPRSS2 (hTMPRSS2) lentiviral vector was purchased from Addgene (pWPI-IRES-Bla-Ak-TMPRSS2; addgene plasmid #154982). *Rhinolophus ferrumequinum* TMPRSS2 (rTMPRSS2) coding sequence was amplified from *Rhinolophus ferrumequinum* cDNA using primers designed to add a N-terminal c-myc Tag and a 5' PmeI restriction site (rTMPRSS2 fwd: 5'-TAGCCTCGAGGTTTAAACCCGGGAGCAGCACCATGGAGCAGAAACTCATCTCTGAAGAG GATCTG ATGGCTTTAAACTCAGGATC) as well as a 3' SpeI restriction site (rTMPRSS2 rev: 5'-GTGGCCACTAG TACGTACGGTCCGCATATGGATCCCTAGCTGTTTGCCCTCATTTGTC). Amplicon were cloned into the backbone of pWPI-IRES-Bla-Ak-TMPRSS2 lentiviral vector using PmeI/SpeI restriction sites. All plasmids were propagated in Stbl3 *E. coli* cells and verified by sequencing (Eurofins Genomics). To generate RFe-AT and RFe-ATC cells, plasmids pTRIP-SFFV-Hygro-2A-TMRPSS2 and pWPI-IRES-Bla-Ak-ACE2 Hygro_TMPRSS2 were used. The lentiviruses coding for the S protein of BANAL-236 and SARS-CoV-2 (BetaCoV/France strain) were previously described [1]. Lentiviral pseudovirions were produced in 293T cells in 10-cm dishes. Co-transfection included the spike plasmid (5 μg), a lentiviral vector expressing luciferase (10 μg, pHAGE-CMV-Luc2-IRES-ZsGreen-W), and auxiliary plasmids (3.3 μg each) HDM-Hgpm2, HDM-tat1b, pRC-CMV-Rev1b using calcium phosphate precipitation. After 5 hours, the medium was replaced with serum-free and phenol red-free DMEM. Pseudo-particles were harvested at 48 hours, clarified by centrifugation, and frozen. Control lentiviruses without S coding sequence were produced in parallel. The amount of pseudo-particles was quantified by p24 ELISA, as described previously [68].

## Cloning of BANAL-236 ORFs

BANAL-236 sequences were amplified from BANAL-236 cDNA using primers designed to add attb1 and attb2 recombination cassettes (S3 Table). Difficult to amplify ORFs were synthetized flanked by attb1 and attb2 sequences. PCR products and synthetic DNA were cloned into pDONOR221entry vector using Gateway BP clonase II Enzyme Mix (Thermo Fisher Scientific). All entry clones were transferred into gateway compatible pciNeo 3X-Flag expression vector (kind gift of Yves Jacob, Institut Pasteur). BANAL-236 NSP3 coding sequence was amplified using primers designed to add 5' NotI and 3' BamHI sites (S2 Table) and in-frame cloned into P3XFLAG-CMV-10 (Promega) using NotI/BamHI restriction enzymes. C-terminally STREP tagged BANAL-236 ORF8 and S expression vector were constructed cloning synthetic sequence

(S3 Table) into pLVX-EF1alpha-IRES-Puro vector (Takara) using EcoRI and BamHI restriction sites. Gene synthesis was performed by Genecust.

Alternatively, SARS-CoV2 and RaTG13 viral open reading frames (ORF) previously cloned in pDONR207 [28,69] were used as templates to derive BANAL-236 sequences using site directed mutagenesis (Quick-change site directed mutagenesis kit, Agilent Technologies Inc.) with primers listed in S4 Table. Previously cloned E, Nsp9, Nsp10 and Nsp16 sequences of SARS-CoV2 in pDONOR207, which are identical to BANAL-236 were not modified.

N-terminally FLAG-tagged BANAL-236 NSP1, NSP13 and ORF6 were amplified using primers designed to add a 5' BamH1 restriction site (BANAL-236-ORF-5'-BamH1_fwd: 5'-TCTAGTGGATCCGCCACCATGGACTACAAAGACCATGA) and a 3' SGRD1 restriction site (BANAL-236-ORF-3'-SGRD1_rev:5'-TGATCCCGTCGACGCCACTTTGTACAAGAAAGCT) using pciNeo 3X-Flag- BANAL-ORFs as a template. Purified amplicons were digested using BamH1/SGRD1 restriction enzymes and subcloned in pFlap-Ubc-nLuc-IRES-Puro lentiviral vector digested with BamHI/XhoI.

## Spike-pseudotyped lentivirus entry assays

Entry assays were conducted with 293T cells stably expressing hACE2-V5, rACE2-V5, myc-hTMPRSS2, myc-rTMPRSS2 or a combination of entry factors [25], BHK-21 cells transiently expressing hACE2-V5, rACE2-V5, myc-hTMPRSS2, myc-rTMPRSS2 or a combination of entry factors, and RFe cells stably expressing hACE2, hTMPRSS2, rACE2, or a combination of entry factors. Cells were transduced in suspension with pseudoviruses at 0.5 ng of HIV-1 p24 antigen for $2.10^4$ cells and then plated in 96-well plates. After 72h of transduction, luciferase activity was measured by adding 100 µl of Bright-Glo substrate (Promega) and quantifying luminescence with a Varioskan LUX luminometer (Thermo Fisher). The expression of the entry factors was assessed by flow cytometry at D0 for each experiment.

## Transfections

Plasmid transfection in 293T was performed using Trans IT-293 (Mirus) following the manufacturer's protocol. For polyI:C stimulation, RFe cells were plated in 12-well plates. The next day, they were transfected with 100 ng polyI:C (InvivoGen) or PBS, respectively, using INTERFERin (Polyplus Transfection) transfection reagent. Caco-2 cells were transfected with 500 ng of polyI:C using lipofectamine RNAiMAX (Thermo Fisher). Cells were lysed 16 h after transfection.

For siRNA transfection, cells were plated in 6-well plates and transfected the next day with 100 nM of siRNA targeting human TMPRSS2 (7113, Horizon Discovery Biosciences), using Lipofectamine RNAiMAX (Thermo Fisher) transfection reagent. Cells were harvested 72 h after transfection.

## Luciferase reporter assays

For luciferase experiments 293T cells were seeded in 96-well plates and transfected using Trans IT-293 (Mirus) or using FuGENE HD (Promega) with a mixture of 20 ng of pISRE-Luc, 2 ng of pRL-TK-Renilla and 20 ng of the plasmids express-ing viral proteins or 20 ng of pCi-Neo plasmids (Promega) as "empty vector" (EV) controls. NS1 Flu (kindly provided by Daniel Marc, INRAE) and NS5 TBEV [47] were used as positives controls. All plasmids were grown in TOP10 cells (Thermo Fisher Scientific) and verified by sequencing. To stimulate the RIG-I/IRF3 axis, 20ng of pdeltaRIG-I (kindly provided by Pierre Genin, Centre d'Immunologie et des Maladies Infectieuses, Paris) was added in the plasmid mix. To activate the JAK/STAT pathway, cells were stimulated with 200 IU/ml of IFNα2b (PBL assay science) 8 hours post-transfection.Twenty-four hours post-transfection, cells were lysed using Passive Lysis buffer (Promega) for at least 15 min and luciferase activity was measured with Dual-Glo Luciferase Assay System (Promega) following the manufacturer's protocol.

## Immunofluorescence microscopy

Cells were fixed with 4% paraformaldehyde (PFA) (Sigma-Aldrich) for 30 min at room temperature. Cells were blocked for 30 min with PBS containing 0.05% Saponin and 5% BSA before incubation with the indicated primary antibodies (S5

Table) for 1 hour. After incubation, cells were washed three times with PBS containing 0.05% saponin and 5% BSA. Alexa Fluor 488 or 647 conjugated secondary antibodies were added for 1 hour. After incubation, cells were washed twice with PBS 0.05% saponin and 5% BSA and once with PBS. Nuclei were stained for 20 min with Hoechst-PBS (Thermo Fisher, 33342 (20 nM)). After washing, slides were mounted with Prolong Gold imaging medium (Life Technologies, P36930). Images were acquired using a Leica SP8 confocal microscope or Zeiss LSM 700 inverted.

## Flow cytometry

Cells were detached with trypsin and fixed in 4% PFA for 30 min at 4°C. Staining was performed in PBS, 2% bovine serum albumin (BSA), 2 mM EDTA (Invitrogen), and 0.1% saponin (FACS buffer). Cells were washed tree time and incubated with indicated antibodies (S5 Table) during 1 hour at 4°C. Cells were washed tree time again and incubated with secondary antibody Alexa 488 or 647 during 45 min in the dark at 4°C. For cell surface staining, cells were detached with trypsin and washed with PBS, 0.5% BSA and 0.4% EDTA (surface buffer). Cells were washed tree time and incubated with indicated antibodies (hACE2-Alexa 647, hTMPRSS2 [30] or anti-N [70]) during 1 hour at 4°C. For hTMPRSS2 staining, cells were washed tree time again and incubated with secondary antibody Alexa 647 during 45 min in the dark at 4°C. Cells were wash tree times again and fixed in 1% PFA for 10 min at 4°C. Data were acquired on an Attune NxT flow cytometer (Thermo Fisher) and were analyzed using FlowJo software v10 (TriStar).

## Transmission Electron Microscopy

About $10^6$ Caco-2 cells were mock-infected or infected with SARS-CoV-2 or BANAL-236 at MOI 0.001 and 0.01 respectively. RFe cells were mock-infected or infected with SARS-CoV-2 or BANAL-236 at MOI of 0.02 and 0.2, respectively. Cells were detached with trypsin and then wash twice with PBS and incubated in 1% glutaraldehyde/4% paraformaldehyde (Sigma, St. Louis, MO) in 0.1 M phosphate buffer (pH 7.2). Samples were then washed in PBS and postfixed by incubation for 1 h with 2% osmium tetroxide (Agar Scientific, Stansted, UK). The cells were then fully dehydrated in a graded series of ethanol solutions and propylene oxide. They were impregnated with a 1:1 mixture of propylene oxide/ Epon resin (Sigma) and left overnight in pure resin. Samples were then embedded in Epon resin (Sigma), which was allowed to polymerize for 48 h at 60°C. Ultrathin sections (90 nm) of these blocks were obtained with a Leica EM UC7 ultramicrotome (Wetzlar, Germany). Sections were stained with 2% uranyl acetate (Agar Scientific) and 5% lead citrate (Sigma), and observations were made with a transmission electron microscope (JEM-1011; JEOL, Tokyo, Japan).

## Western blot analysis

Cells were lysed in RIPA buffer (Sigma-Aldrich) supplemented with a protease and phosphatase inhibitor cocktail (Roche). Samples were denatured in 4X loading buffer (Li-Cor Bioscience) under reducing conditions (NuPAGE reducing agent, Thermo Fisher Scientific). Proteins were separated by SDS-PAGE (NuPAGE 4–12% Bis-Tris gel or NuPAGE 10–20% Tricine gel for small protein, Invitrogen) in running buffer MOPS (Invitrogen) or MES (Invitrogen) for small proteins. Proteins were transferred to nitrocellulose membranes (Bio-Rad) using a Trans-Blot Turbo system (Bio-Rad) or liquid transfer for NSP3. Membranes were blocked with PBS-Tween 0.1% containing 5% milk. After blocking, membranes were incubated overnight at 4°C with primary antibodies (S5 Table) diluted in blocking buffer or PBS-Tween 0.1%. Finally, membranes were washed and incubated for 45 minutes at room temperature with diluted secondary antibodies, then washed three time again. Images were acquired with an Odyssey CLx infrared imaging system (Li-Cor Bioscience).

## Antibodies and cytokines

Details about antibodies are given in S5 Table. Cells were stimulated overnight with IFNα2b (PBL Biosciences) at a final concentration of 1 000 IU/ml.

## RNA extractions and RT-qPCR assays

Total RNA was extracted from cell lysates using the NucleoSpin RNA II kit (Macherey-Nagel) and eluted in water. cDNA synthesis was performed on 1 µg of total RNA with RevertAid H Minus M-MuLV reverse transcriptase (Thermo Fisher Scientific) using random p(dN)6 primers (Roche). Real-time quantitative PCR was carried out on a Quant Studio 6 Flex system (Applied Biosystems) with SYBR green PCR mix (Life Technologies). Data were analyzed using the ΔΔCT method, with all samples normalized to GAPDH. All experiments were performed in technical triplicate. The primers used for RT-qPCR analysis are listed in S6 Table. Genome equivalent concentrations were determined by extrapolation from a standard curve generated from serial dilutions of plasmids expressing hACE2-V5, myc-hTMPRSS2, rACE2-V5, myc-rTMPRSS2 or BANAL-236 E protein.

## RNA library and sequencing

Library preparation and sequencing was performed by the Biomics platform at the Institut Pasteur. cDNA libraries were prepared from 100-200 ng RNA using a Illumina Stranded mRNA library Preparation Kit (Illumina, USA) following the manufacturer's protocol whereby adaptors are added to cDNA by A-tail mediated ligation. Index barcodes were added by PCR of 16 cycles. Unbound adaptors and indexes were eliminated by purification on AMPure magnetic beads (Beckman-Coulter). The resulting library featured an electrophoretic profile of 200–1000 bp, with a major peak of 375 bp, as visualized on a 3500 Fragment Analyzer (Agilent). A NextSeq 2000 sequencing system and a P4 50c flowcell (Illumina) were used to obtain 67-nt single-end double-indexed reads.

## RNA-seq analysis

The RNA-seq analysis was performed with Sequana 0.18.0 [71]. We used the RNA-seq pipeline 0.20.0 (https://github.com/sequana/sequana_rnaseq) built on top of Snakemake 7.32.4 [72]. Briefly, reads were trimmed from adapters using Fastp 0.23.2 [73] then mapped to the corresponding genome using STAR 2.7.10a [74]. *Rhinolophus ferrumequinum* RFe-AT and RFe-TC samples were mapped to GCF_004115265.2 assembly with annotation GCF_004115265.2-RS_2023_02 from NCBI. FeatureCounts 2.0.1 [75] was used to produce the count matrix, assigning reads to features using corresponding annotation with strand-specificity information. Quality control statistics were summarized using MultiQC 1.17 [76]. Statistical analysis on the count matrix was performed to identify differentially regulated genes. Clustering of transcriptomic profiles were assessed using a Principal Component Analysis (PCA). Differential expression testing on each strain separately was conducted using DESeq2 library 1.38.3 [77] scripts indicating the significance (Benjamini-Hochberg adjusted p-values, false discovery rate FDR < 0.05) and the effect size (fold-change) for each comparison. DESeq2 output files (S1 Table) was used to filter and plot Differentially Expressed Genes (DEG) (Log2 fold change ≥ 1.5, adjusted p-value < 0.05). The Gene Ontology GO enrichment (Biological process) on the list of differently upregulated gene was analyzed using Mestascape [78].

**Sequencing of BANAL-236 stocks** was performed as previously [6] except that the SMARTer Stranded Total RNA-Seq Kit v3 – Pico Input Mammalian library was sequenced on an Illumina NextSeq2000 in a paired-end 200-cycle run. Raw reads were trimmed with AlienTrimmer and further mapped onto BANAL-236 genome (GenBank accession number MZ9370003.2) using CLC Genomics Workbench 25.0.2 with read Length fraction = 1 and Similarity fraction = 0.985. Low Frequency Variant Detection was performed with CLC Genomics Workbench 25.0.2 with the following parameters: required significance >1%, minimum read coverage 100, minimum read count 10, minimum frequency 1%, minimum central quality 20, minimum neighborhood quality 15, relative read direction filter 1%, removal of pyro-error variants. Filtering of variants hits detected by CLC were manually performed with the following parameters: minimum base quality 30, forward/reverse balance >0. Variants selected after this filtering were finally checked for position on read to remove variants positioned on read ends. Mutations harbouring an allelic frequency above 1% were selected (S7 Table) while mutations with low frequency and weak variation were discarded as putative sequencing error artifacts.

## Data analysis

Data are presented and analyzed using GraphPad Prism 9. Alignments and trees were generated using CLC Genomics Workbench 22. Immunofluorescence quantification was done using Zen Blue software.

## Supporting information

**S1 Fig. The tagged versions of ACE2 (ACE2-V5) and TMPRSS2 (Myc-TMPRSS2) are well expressed and functional.**
(PDF)

**S2 Fig. Investigations of BANAL-236 entry mechanisms in 293T, BHK-21 and RFe cells.**
(PDF)

**S3 Fig. Transmission electron microscopy analysis of Caco-2 and RFe-ATC cells.**
(PDF)

**S4 Fig. Interferon-stimulated genes are not induced upon SARS-CoV-2 and BANAL-236 replication in Caco-2 and RFe-ATC cells.**
(PDF)

**S5 Fig. RFe-AT and RFe-ATC cells are immunocompetent.**
(PDF)

**S1 Table. Transcriptomic analysis of RFe-AT and RFe-ATC cells, stimulated with polyI:C or left unstimulated.**
(XLSX)

**S2 Table. Number of residues that differ between BANAL-236 and SARS-CoV-2 (Wuhan) proteins.**
(PPTX)

**S3 Table. Primers and sequences used for cloning BANAL-236 ORFs.**
(DOCX)

**S4 Table. Primers used for mutagenesis.**
(DOCX)

**S5 Table. Antibodies used in the study.**
(DOCX)

**S6 Table. Primers used for qPCR amplification of Human and *Rhinolophus ferrumequinum* genes.**
(DOCX)

**S7 Table. Frequency of allelic mutations observed in BANAL-236 C3 passage performed in VeroE6 cells.**
(DOCX)

## Acknowledgments

We thank the members of the Virus and Immunity Unit at the Institut Pasteur in Paris (Andréa Cottignies-Calamarte, Françoise Porrot, Florence Guivel-Benhassine, Julian Buchrieser, Nell Saunders, Mariem Znaidia, Inès Maréchal, and Nicoletta Casartelli) for their generous provision of reagents, expert advice, p24 titration, and discussions. We are grateful to the French National Reference Centre for Respiratory Viruses hosted by Institut Pasteur (Paris, France) and led at the time by Pr. S. van der Werf, for providing SARS-CoV-2 and BANAL-236 strains. We also extend our thanks to Adrià

Sogues for insightful AF3-related discussions, to Sophie-Marie Aicher and Amandine Chantharath for their help initiating this project, and Audrey Salles from the Photonic BioImaging platform of the Institut Pasteur for her help and advice. Finally, we thank Etienne Kornobis and Iakov Vitrenko from the Biomics platform (C2RT, Institut Pasteur, Paris), which is supported by France Génomique (ANR-10-INBS-09) and IBISA, for performing the transcriptomic analyses.

## Author contributions

**Conceptualization:** Ségolène Gracias, Elodie Le Seac'h, Françoise Vuillier, Sarah Temmam, Sandie Munier, Adolfo Garcia-Sastre, Vincent Caval, Nolwenn Jouvenet.

**Data curation:** Ségolène Gracias, Elodie Le Seac'h, Françoise Vuillier, Léa Vendramini, Adam Moundib, Sarah Temmam, Vincent Caval, Nolwenn Jouvenet.

**Formal analysis:** Ségolène Gracias, Elodie Le Seac'h, Léa Vendramini, Adam Moundib, Sarah Temmam, Philippe Roingeard, Vincent Caval, Nolwenn Jouvenet.

**Funding acquisition:** Olivier SCHWARTZ, Nevan J Krogan, Caroline Demeret, Adolfo Garcia-Sastre, Nolwenn Jouvenet.

**Investigation:** Ségolène Gracias, Elodie Le Seac'h, Samuel Donaire-Carpio, Françoise Vuillier, Léa Vendramini, Adam Moundib, Sarah Temmam, Flora Donati, Philippe Roingeard, Jyoti Batra, Vincent Caval.

**Methodology:** Ségolène Gracias, Elodie Le Seac'h, Samuel Donaire-Carpio, Françoise Vuillier, Adam Moundib, Sarah Temmam, Flora Donati, Sandie Munier, Philippe Roingeard, Jyoti Batra, Vincent Caval.

**Project administration:** Nevan J Krogan, Adolfo Garcia-Sastre, Vincent Caval, Nolwenn Jouvenet.

**Resources:** Samuel Donaire-Carpio, Magdalena Rutkowska, Flora Donati, Anastasija Cupic, Javier Juste, Carles Martinez-Romero, Nathalie Morel, Olivier SCHWARTZ, Nevan J Krogan, Lisa Miorin, Marcel A Müller, Caroline Demeret, Adolfo Garcia-Sastre, Vincent Caval.

**Supervision:** Ségolène Gracias, Nevan J Krogan, Lisa Miorin, Caroline Demeret, Sandie Munier, Adolfo Garcia-Sastre, Vincent Caval, Nolwenn Jouvenet.

**Validation:** Ségolène Gracias, Elodie Le Seac'h, Sandie Munier, Philippe Roingeard, Vincent Caval, Nolwenn Jouvenet.

**Visualization:** Ségolène Gracias, Elodie Le Seac'h, Vincent Caval, Nolwenn Jouvenet.

**Writing – original draft:** Ségolène Gracias, Elodie Le Seac'h, Vincent Caval, Nolwenn Jouvenet.

**Writing – review & editing:** Ségolène Gracias, Elodie Le Seac'h, Samuel Donaire-Carpio, Françoise Vuillier, Léa Vendramini, Adam Moundib, Sarah Temmam, Magdalena Rutkowska, Jyoti Batra, Vincent Caval, Nolwenn Jouvenet.

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
