## [Decision Letter · Decision Letter 0]

26 Oct 2025

Entry, replication and innate immunity evasion of BANAL-236, a SARS-CoV-2-related bat virus, in Rhinolophus and human cells

PLOS Pathogens

Dear Dr. Jouvenet,

Thank you for submitting your manuscript to PLOS Pathogens. After careful consideration, we feel that it has merit but does not fully meet PLOS Pathogens's publication criteria as it currently stands. Therefore, we invite you to submit a revised version of the manuscript that addresses the points raised during the review process.

Please submit your revised manuscript within 60 days Dec 25 2025 11:59PM. If you will need more time than this to complete your revisions, please reply to this message or contact the journal office at plospathogens@plos.org. Please include the following items when submitting your revised manuscript:

We look forward to receiving your revised manuscript.

Kind regards,

Jesse D Bloom, Ph.D.

Guest Editor

PLOS Pathogens

Alexander Gorbalenya

Section Editor

Editor-in-Chief

PLOS Pathogens

PLOS Pathogens

orcid.org/0000-0002-7699-2064

**Additional Editor Comments:**

EDITOR COMMENTS:

I obtained two reviews from experts, and have also read the manuscript carefully myself. The reviewers had many critiques, most of which I concur with. This paper does have a lot of interesting findings including looking at aspects of replication in bat versus human cells, but many of the claims are overstated. I am willing to consider a revised version of the manuscript, but only if it makes large-scale changes to address the reviewer comments as well as my own comments which I have put directly below. This will require substantial revision, and please only submit a revised version if you can address the main points and much more clearly not make over-claims related to points that cannot be fully addressed.

MAJOR COMMENTS:

- The manuscript contains numerous statements suggesting that BANAL-236 is more efficient at using human ACE2 than Rhinolophus ACE2. As also pointed out by reviewer 1, these statements are not justified and given the wrong impression. Rhinolophus bat ACE2 is known to be under positive selection for variation in the region bound by sarbecovirus spikes, such that different Rhinolophus alleles can have dramatically different abilities to support cell entry (eg, see Guo et al, Journal Virology, 94:e00902-20; Starr et al, Nature, 603:913; Li et al, PLoS Pathogens, 10.1371/journal.ppat.1011116; and many other papers). Furthermore, there is no reason to think that sequencing of Rhinolophus bat ACE2s is anywhere close to exhaustively sampling all the natural alleles, and further this paper did not test most of the known alleles. Therefore, to imply that BANAL-236 uses human ACE2 more efficiently than Rhinolophus ACE2 is not supported without more exhaustively testing Rhinolophus ACE2s. So the paper should not claim as a conclusion anything about ACE2 usage, rather this is more like an experimental limitation that only limited testing of different Rhinolophus ACE2 alleles was done.

- What controls were performed to ensure that the epitope tags introduced onto the bat ACE2 and TMPRSS2 did not interfere with their functions as receptors and spike activating proteases?

- Overall, as noted by both reviewers, while this study has interesting findings the limitations associated with using a single Rhinolophus cell line to make overly broad claims should be better described throughout the manuscript.

MINOR COMMENTS:

- Line 42: might clarify that you mean isolated directly from a primary sample, as other related viruses have been made by reverse genetics.

- Abstract lines 62-64: says it is "essential" to study BANAL-236 replication in bat cells to identify factors that could affect zoonotic spillover. This does not really make sense: it's not clear how studying in bat cells is essential to identify mechanisms that facilitate zoonotic spillover to humans; studying in bat cells is important for understanding replication in the natural host. I'm not disagreeing that this study is interesting or important, but this statement does not seem supported so remove this overstated rationale.

- Line 77-78 of Introduction: the SARS-CoV-2 genome is highly recombinant with different closest relatives in different genomic regions, so rather than saying the viruses are "the closest known relatives" it should say something more like "are among the closest known relatives" as for instance there are parts of the genome where RmYN02 (and other viruses) are more similar. Likewise, in the next sentence SARS-CoV-2 is unlikely to have arisen through recombination of these exact viruses, it would be better to say more like "An ancestor of SARS-CoV-2 may have originated from recombination and evolution of close relatives of these viruses and other Rhinolophus sarbecoviruses."

- Line 84-85: should probably note that all other sarbecoviruses (not just the BANAL ones) lack the FCS.

Line 154-155: says replication of BANAL-236 similar to SARS-CoV-2 in Caco-2 cells, but in fact replication clearly worse: more virus had to be added, increase in titer from first to second timepoint was slower, and fewer cells every became N positive. In fact, the rest of the paragraph makes exactly this point, so it's not clear why this sentence says the replication kinetics are similar when in fact they are clearly slower.

**Journal Requirements:**

At this stage, the following Authors/Authors require contributions: Ségolène Gracias, Samuel Donaire-Carpio, Françoise Vuillier, Elodie Le Seac’h, Léa Vendramini, Adam Moundib, Magdalena Rutkowska, Anastasija Cupic, Javier Juste, Sarah Temmam, Flora Donati, Carles Martinez-Romero, Nathalie Morel, Olivier SCHWARTZ, Nevan Krogan, Lisa Miorin, Caroline Demeret, Philippe Roingeard, Sandie Munier, Jyoti Batra, Adolfo Garcia-Sastre, vincent Caval, and Nolwenn Jouvenet. Please ensure that the full contributions of each author are acknowledged in the "Add/Edit/Remove Authors" section of our submission form.

- ® on page: 16.

4) Please amend your detailed Financial Disclosure statement. This is published with the article. It must therefore be completed in full sentences and contain the exact wording you wish to be published.

5) Please provide separate figure files in .tif or .eps format.

For more information about figure files please see our guidelines

https://journals.plos.org/plospathogens/s/figures#loc-file-requirements

6) Please send a completed 'Competing Interests' statement, including any COIs declared by your co-authors. If you have no competing interests to declare, please state "The authors have declared that no competing interests exist". Otherwise please declare all competing interests beginning with the statement "I have read the journal's policy and the authors of this manuscript have the following competing interests:"

**Reviewers' Comments:**

Reviewer's Responses to Questions

**Part I - Summary**

Reviewer #1: Recent virus discovery studies have identified close relatives of SARS-CoV-2 in bats across southeast Asia. BANAL-236 and BANL-52 are two viruses identified in R. marshalli and R.malayanus, respectively. In this study, Gracias and colleagues performed a series of in vitro based experiments characterizing receptor compatibility, cell entry and viral replication of these viruses in bat and human cell lines. The results show curiously limited bat receptor use, strong human receptor compatibility, and the ability of BANAL-236 to partially inhibit interferon responses. While the results are interesting and help provide insights into interferon antagonism by SARS-CoV-2-related bat viruses, this study lightly hand waves a fair amount of bat biology at the expense of more meaningful conclusions. Regardless of these technical limitations (they can be easily addressed), the findings are timely and of interest to the PLoS Pathogens audience.

Reviewer #2: Gracias et. al characterize the replication and innate immune evasion of BANAL-236, a SARS-CoV-2-related bat sarbecovirus in both human and a novel Rhinolophus fibroblast bat cell line. Using both full-length and pseudotyped BANAL-236 virus, the authors demonstrate that BANAL-236 preferentially uses human ACE2 and TMPRSS2 over bat orthologs, and antagonizes host innate immune response pathways, effectively highlighting the zoonotic potential of this SARS-CoV-2-related coronavirus. Considering the ability of these viruses to recombine in nature and potentially adapt to human hosts, the study is timely and of interest to the coronavirus community. Experiments are also excellently conducted and the quality of the data is superb. However, there is a lack of mechanistic depth in describing the innate immune evasion pathways employed by BANAL-236, and the clonally derived bat cell line model is limited in its overall utility for studying these pathways. There are several significant gaps in the study, and their absence diminishes the overall impact of this manuscript and it’s fit within this journal. Major revisions are required to substantiate the overall claims.

**Part II – Major Issues: Key Experiments Required for Acceptance**

Please use this section to detail the key new experiments or modifications of existing experiments that should be absolutely required to validate study conclusions.required to validate study conclusions.required to validate study conclusions.required to validate study conclusions.

Reviewer #1: MAJOR:

1. BANAL-236 was identified in R. marshalli and BANAL-52 was identified in R.malayanus. While convenient for its accessibility, R. ferrumequinum is not even from the same continent. I understand cells and tissues from the true BANAL virus hosts are not readily available, but the text should at least highlight this limitation when interpreting the results.

2. All of the BANAL viruses were identified in fecal samples, and basically every other bat sarbecovirus described to date was identified in rectal/anal swabs or fecal samples. Therefore, I do not understand the authors' decision to produce lung-derived fibroblast cell lines to study these viruses. The authors claim the Rfe lung fibroblast cells are a relevant model but given that this is not the correct host species or presumed target tissue, these claims on "relevance" should be tempered. Maybe these bat cells are more relevant than 293Ts, but then again, 293Ts and other standard lab lines are used in this study to show many of the points because the RFe cells have issues.

3. Several recent studies have demonstrated the majority of beta-coronaviruses exhibit fairly tight species specificity (PMID: 39402048, 35114688, 40436893, 35895696, 40934083, 20567988), and even sarbecoviruses identified in R. sinicus are not capable of using every ACE2 isoform from R. sinicus bats (PMID: 32699095, 37655938). The 293T-based over-expresion experiment in figure 2E shows that BANAL-52 can use ACE2 from its own species while BANAL-236 does not use the same ACE2 orthologue, further supporting the idea that BANAL viruses only poorly use other bat ACE2s. This manuscript then claims BANAL-236 and -52 use hACE2 "more efficiently" than from their own species (for example, lines 48-49, 200-203, 386-388) and lines 397-398 sound like other studies that have failed to use the appropriate host species of receptor and then speculate on "alternative receptor" use (for example, PMID: 36477529 claims HKU5 may use an alternative receptor because it did not interact with a variant of ACE2 from a closely related bat species than what the virus was identified in- later shown to be false). In general, these claims should be adjusted to highlight how the appropriate ACE2 orthologue was not actually tested for -236. Maybe AlphaFold3 could be used to perform some in silico predictions.

4. While R. mashalli ACE2 is not publicly available, the authors DO have a species-matched ACE2 with BANAL-52, which they could study further to strengthen their claims. For instance, I believe these experiments could be replicated in alternative cell backgrounds, such as BHK, Huh7.5, or Veros, to assess the consistency of the results. Although BANAL-52 appears to use R. malayanus ACE2 "less" than human ACE2 in figure 2E, this experiment was performed in 293T cells. For the human side of this experiment, the authors use human ACE2, human TMPRSS2 and human cells but for the bat side they are using R. malayanus ACE2, R. ferrumequinum TMPRSS2 and human cells. Can the authors speculate (or test) if these mixed bat factors in human cells are all properly co-localizing on the membrane as efficiently as all of the human entry factors? It may be possible that human ACE2 interacts with other human factors/microdomains involved in entry better than bat ACE2, so it seems difficult to really conclude from this one experiment that the bat virus RBD affinity for human ACE2 is really stronger.

5. BANAL-236 was obtained from a third party for this study and further passaged in Veros but there is no mention of additional whole viral sequencing to verify a pure stock without cell culture adaptations. Because Vero adaptation is very common for cornaviruses, the original BANAL-236 study isolated the virus in Veros and sequence verified after few passages to show it did not adapt to Veros. Therefore, further passaging this virus in Vero cells, as was done in this study, warrants further sequencing.

6. Figure 6A: The RFe-ATC cells appear to be more interferon deficient than the parental cells. Is it possible the authors have selected a slightly less immunocompetent cell line, accounting for the differences in replication phenotype in figure 3?

7. I really liked the last experiment in figure 7 with the different viral genes. Nsp1, Nsp13 and Orf6 could be further tested in the RFe cells to see if they modulate the responses measured in figure 6.

Reviewer #2: 1.) Rigor of the BANAL-236 virus model. Interestingly, BANAL-236 is reported to use human ACE2 for cellular entry with higher efficiency compared to the ACE2 receptor from its cognate host. To strengthen this claim, it would be helpful if the authors could provide more details about how viral stocks were prepared. How many times was the virus passaged to generate working stocks used in these studies? Was the stock deep sequenced to rule out cell culture adaptive mutations (i.e., mutations that enhance affinity for African green monkey ACE2 since the virus was grown in VeroE6 cells)? There is currently no mention of these methods within the manuscript.

2.) Utility of the Rhinolophus fibroblast cellular model. The authors developed and characterized a novel Rhinolophus bat lung fibroblast cell model. While this presents a valuable step toward expanding bat-derived experimental systems (which are currently lacking in the field), the demonstrated utility of the model is limited. Only a single clonally derived cell line expressing human ACE2 and TMPRSS2 (RFe-ATC) were found to be susceptible to BANAL-236, and this clone was artificially engineered to express human ACE2 and TMPRSS2. Because BANAL-236 susceptibility depends on exogenous expression of human, rather than an endogenous Rhinolophus factor(s), the relevance is unclear and overall enthusiasm for this cellular model is diminished.

3.) Mechanistic depth describing innate immune evasion pathways. Antagonism of host innate immune responses is well established for human and zoonotic coronaviruses, and these molecular mechanisms are well described. The reviewer appreciates the tremendous effort and experimental rigor in defining the innate immune antagonistic ability of BANAL-236 and the subsequent identification of the viral proteins involved in this process. However, considering the level of mechanistic detail already established for how human and zoonotic CoVs antagonize host innate immune pathways, the current study does not substantively contribute to our overall knowledge of these pathways. Additionally, only three interferon stimulated genes (OAS1, ISG20, and IFITM3) are considered in the evaluation BANAL-236’s immune evasion. Further transcriptomic characterization is needed to define the innate immune response the RFeATC cell line. Furthermore, experiments examining the role of specific viral proteins in innate immune antagonism are unfortunately only conducted in HEK293T cell due to technical considerations. This again limits the relevance of the RFe-ATC model and weakens claims regarding the ability of BANAL-236 to evade bat innate immune responses.

**Part III – Minor Issues: Editorial and Data Presentation Modifications**

Reviewer #1: MINOR

1. Cell surface expression of bat ACE2 is measured by flow cytometry using human ACE2 antibodies. It might also be nice to show overall cellular protein levels by westernblot. Tagged receptors could help reduce non-specific effects from differences in ACE2 antibody cross-reactivity.

2. Related to minor point 1, the quantitative PCR does not seem to use primers specific for R. ferrumequinum genes. How well do the human primers work with the bat genes?

3. Related to major points 3 and 4, were the bat virus genes codon optimized for human cells?

4. Lentiviruses can fail to faithfully recapitulate bat sarebocovirus cell entry (PMID: 26552008, supp. figure 1D). Using an alternative system like VSV may provide additional insights.

5. The authors have issues maintaining expression in the bat cells and could not make a line expressing both bat ACE2 and bat TMPRSS2. What lentivector is being used? It seems like multiple promoters are being used instead of using IRES or 2A sites, which is likely driving at least some of these challenges.

6. Figure 6A and B: the colored dots are not easy to see against the gray bars. The bars could also be colored similarly.

7. The title for supplemental figure 3 appears to have a find and replace error from adjusting the name of the cell line: "InteRFeron"

8. VSV could be an interesting virus to use in the bat cells to show interferon response to infection. It is fairly immuno-stimulatory in many cell lines.

Reviewer #2: 1.) The author’s report that RFe-ATC cells are difficult to transfect. Have they tried additional transfection reagents shown to be effective for otherwise hard-to-transfect cell lines? If so, please mention.

2.) The use of both full-length virus and pseudotyped viruses strengthens the rigor of the work. However, the figure legends are not immediately apparent to the reader which platform was used in each experiment.

PLOS authors have the option to publish the peer review history of their article (what does this mean?). If published, this will include your full peer review and any attached files.). If published, this will include your full peer review and any attached files.). If published, this will include your full peer review and any attached files.). If published, this will include your full peer review and any attached files.

...

Reviewer #1: No

Reviewer #2: No

**Figure resubmission:**

**Reproducibility:**



---

## [Decision Letter · Decision Letter 1]

31 Mar 2026

Dear Dr Jouvenet,

We are pleased to inform you that your manuscript 'Entry, replication and innate immunity evasion of BANAL-236, a SARS-CoV-2-related bat virus, in Rhinolophus and human cells' has been provisionally accepted for publication in PLOS Pathogens (see Editor's comments below)

Best regards,

Jesse D Bloom, Ph.D.

Guest Editor

PLOS Pathogens

Alexander Gorbalenya

Section Editor

PLOS Pathogens

Sumita Bhaduri-McIntosh

Editor-in-Chief

PLOS Pathogens

orcid.org/0000-0003-2946-9497

Michael Malim

Editor-in-Chief

PLOS Pathogens

orcid.org/0000-0002-7699-2064

Thanks for the thorough revisions of your manuscript. Both reviewers (and I) agree that you did a very complete job of addressing the critiques, and congratulations on the nice study.

Do be sure (as noted by Reviewer 2) to upload the raw sequencing date (eg, RNA-seq) to the SRA, ENA, or GEO (eg, one of the INSDC databases). I am recommending acceptance on the condition that data upload is done.

Reviewer Comments (if any, and for reference):

Reviewer's Responses to Questions

**Part I - Summary**

Reviewer #1: The authors have taken provided a very thorough rebuttal addressing all of my comments.

Reviewer #2: The authors have thoroughly addressed my concerns and significantly improved the overall impact of the paper. The RNAseq experiment was well done and an excellent addition to the paper.

**Part II – Major Issues: Key Experiments Required for Acceptance**

Please use this section to detail the key new experiments or modifications of existing experiments that should be absolutely required to validate study conclusions.required to validate study conclusions.required to validate study conclusions.required to validate study conclusions.

Reviewer #1: none

Reviewer #2: (No Response)

**Part III – Minor Issues: Editorial and Data Presentation Modifications**

Reviewer #1: none

Reviewer #2: Will the raw RNAseq data be made publicly available? If so, please upload to SRA/GEO and provide accession numbers to these datasets.

PLOS authors have the option to publish the peer review history of their article (what does this mean?). If published, this will include your full peer review and any attached files.). If published, this will include your full peer review and any attached files.). If published, this will include your full peer review and any attached files.). If published, this will include your full peer review and any attached files.

...

Reviewer #1: No

Reviewer #2: No

---

## [Editor Report · Acceptance letter]

Dear Dr Jouvenet,

We are delighted to inform you that your manuscript, "Entry, replication and innate immunity evasion of BANAL-236, a SARS-CoV-2-related bat virus, in Rhinolophus  and human cells," has been formally accepted for publication in PLOS Pathogens.

Best regards,

Sumita Bhaduri-McIntosh

Editor-in-Chief

PLOS Pathogens

orcid.org/0000-0003-2946-9497

Michael Malim

Editor-in-Chief

PLOS Pathogens

orcid.org/0000-0002-7699-2064